# A Load-Balance System Design of Microgrid Cluster Based on Hierarchical Petri Nets

**Jose R. Sicchar** [1,*], **Carlos T. Da Costa Jr.** [2], **Jose R. Silva** [3], **Raimundo C. Oliveira** [4] **and Werbeston D. Oliveira** [5]

1   Control and Automation Engineering Department, High School Technology,
    University of the Amazon State, Manaus 69050-025, Brazil
2   Electrical Engineering Faculty, Institute of Technology, Federal University of Pará,
    Belém 66075-110, Brazil; cartav@ufpa.br
3   Mechatronic Department, Polytechnic High School, University of São Paulo,
    São Paulo 05508-900, Brazil; reinaldo@usp.br
4   Computation Engineering Department, University of the Amazon State, Manaus 69050-025, Brazil;
    rcoliveira@uea.edu.br
5   Electric Engineering Department, Federal University of Amapá,
    Macapá, Amapá 68903-419, Brazil; wdoliveira@unifap.br
*   Correspondence: jvilchez@uea.edu.br; Tel.: +55-92-99221-2832

**Abstract:** In the new paradigm of urban microgrids, load-balancing control becomes essential to ensure the balance and quality of energy consumption. Thus, phase-load balance method becomes an alternative solution in the absence of distributed generation sources. Development of efficient and robust load-balancing control algorithms becomes useful for guaranteeing the load balance between phases and consumers, as well as to establish an automatic integration between the secondary grid and the supervisory center. This article presents a new phase-balancing control model based on hierarchical Petri nets (PNs) to encapsulate procedures and subroutines, and to verify the properties of a combined algorithm system, identifying the load imbalance in phases and improving the selection process of single-phase consumer units for switching, which is based on load-imbalance level and its future state of load consumption. A reliable flow of automated procedures is obtained, which effectively guarantees the load equalization in the low-voltage grid.

**Keywords:** hierarchical Petri nets; urban microgrids; phase-load balancing

---

## 1. Introduction

Electric energy distribution in low voltage (LV) can be enhanced by a distributed architecture based on *urban microgrids*(UMG) [1–4]. A microgrid is essentially a cluster of residential consumers where at least some consumers possess local energy sources and a storage system. Energy supply in this system is a balance between electric power provided by a power line and that obtained from domestic loads generated by user sources [5–7]. Supervision and control of energy flow is managed by a **Microgrid Central Control**(MGCC) [8–10], which manages the balance between energy consumption, the main supply, and energy from microgrid components [11–14].

Nevertheless, in the legacy LV system [15] the phase-load imbalance is a drawback, especially because domestic loads generated by single-phase consumers affect grid phase stability, and the energy quality supplied [16,17]. Thus, some methods of solving this problem are highlighted in the the electrical current injection from distributed generation microgrids [18–20], the coordinated load balance [16,21], the integrated multimicrogrid control [12,22–24] and the load phase balance [25–27].

In the case of urban microgrids with distributed generation, the load-balancing method is based on the "electric current injection" in consumer unit phases, as well as in the phases of the LV grid, compensating for the imbalance of load and voltage. However, it is necessary to use a complex AC/DC–DC/AC signal converter control architecture called Microgrid Central Control (MGCC) [28], frequency inverters [29] and, in particular, supervision and control algorithms that optimize power and electric current flow [20]. The MGCC usually manages this automated solution flow, which does not always guarantee the efficient control of the phase shift effects between the main electrical current and the injected electrical current [30].

The load-balance procedure based on the "coordinated load balance" offers a wide range of control features for current injection, working synchronously with the grid transformer [16], with frequency compensation between the grid phases and consumer units, along with phase compensation between the grids' electrical current and the electric current injected [31]. Ensuring robustness and load balancing, however, requires a complex central control and supervision structure with local (distributed) controllers with high-reliability algorithms [32] that ensure automated operational integration at all control and supervisory levels.

Another method of load balance based on "integrated multimicrogrids control" is being widely used because of the large mix of micro-sources of energy to be applied for load-balancing [22,29,33], along with frequency and phase compensation in the grid and consumer units [34], also requiring a complex architecture with control and supervision algorithms that efficiently coordinate current injection and frequency and phase compensation in the LV grid [9,11,35], as well such as a large number of distributed generation units [36], which in fact means a great limitation for a large-scale implementation in developing countries [7,37].

An alternative to implementing the above-mentioned techniques is phase-load balancing, which consists of switching single-phase consumer units to the phases of the LV grid that are balanced. The procedure is based on the use of identification algorithms and load transfer management, aiming at minimizing current and load consumption [38] or voltage and load [27]. In both cases, the voltage and load equilibrium state in the grid phases is guaranteed; however, the switching choice is based only on current load consumption of consumers' units, disregarding the imbalance level and the future states of load consumption, which could contribute to the robustness of the system to eventual consumption peaks and to the durability of the load stability over time.

By contrast, it has been observed that the use of Petri nets (PNs) in complex systems is quite broad [39], due to its formal modeling, simulation and property verification capabilities [40–42], which allows development and verification of intelligent algorithms for control and supervision of application in smart grids [43,44]. The formal verification of routine flow allows evaluation of incidences, conflicts, deadlocks, loops, and reachability [45] of all stages and subroutines, as well as evaluation of inviolable flows and cycles of the algorithm in all its hierarchical levels [46], and also the automatic integration workflow with the control and supervision systems of an urban microgrid [47].

Thus, the use of PNs can contribute to the solution of the lack of automation in the operational procedures of load balancing in urban microgrids and especially in the LV grid [48], such as in the case of the legacy Brazilian LV distribution grid [15], with partially automated flows and manual methods without automatic full flow with the central supervisory system. Therefore, the existence of an intelligent algorithm that allows automation of the load-balancing procedures in the LV grid, as well as automatic integration at all levels of the grid control and supervision, would generate a great improvement in the legacy methods of load balancing, with correct, reliable and efficient processes, guaranteeing the load stability, as well as the streamlining of operational procedures in case of possible problems of load and voltage imbalance in the grid, and even emergency situations such as the burning of the transformer, among others.

In this article, we present a new system design, based on hierarchical PNs, of an intelligent algorithm to automate the load-balancing process, in order to provide reliable and effective procedures and to integrate efficiently the automation workflow in the legacy Brazilian LV grid.

The authors believe that the main contribution of this paper lies in providing a formal process-automation model that optimizes and integrates the workflow of a load-balancing control system in the legacy LV grid. The proposed control system is based on combined algorithms to minimize load consumption in the grid phases (feeders), through following programmable procedures: "load transfer in the grid feeders", which is based on a fuzzy inference to identify and perform the load transfer or between feeders; "imbalanced consumer unit identification", which is based on a fuzzy inference system to detect load-imbalance level in consumer units; "load forecast in consumers units", which is based on a Markov chain algorithm that forecasts monthly levels of discrete states on load consumption; and "switch selection" which is based on an optimal choice algorithm of imbalanced consumer unit with high load consumption.

The main contribution of this paper can be summarized as follows:

- A novel system design of a load-balance system integrated with the legacy LV system and urban microgrids is proposed. This is validated in Petri nets, emphasizing the novel form of encapsulating combined algorithms, evidenced by hierarchy levels of integration [43];
- The reachability graph and place-invariant analysis for property verification and the experimental assessment of robustness and efficiency of the load-balance algorithm is used. In addition, simulation dynamic tests are applied in a real case study of a LV grid of a city in the north of Brazil, for performance analyses of the proposed algorithm. Stored data about user consumption and grid feeders were used for simulation and analysis.
- A new method of choosing single-phase consumer units for the load-balancing process based on the imbalance levels and future states of load consumption, resulting in the efficient attenuation of the load average imbalance between LV feeders is applied, in comparison to the legacy system method and the bibliographic revision, which consider in both cases only the current load consumption.

The proposed system was validated efficiently through obtained results, providing an efficient and an automated reliable workflow for the load-balancing process in the legacy LV grid, which may also became an alternative load-balancing control procedure at the MGCC, in the urban microgrid context to operate as a coordinated control system with the current injection system of microgrids.

The remainder of this article is organized as follows: Section 2 explains the related background. In Section 3 the load-balancing control architecture is presented. In Section 4 the experimental system design validation and simulation dynamic results are presented and discussed. Finally, the conclusions and suggestions of this study are presented.

## 2. Background

In this section, we address some related issues that support the proposal. First, we present the state of art regarding urban microgrids. Next we address the load-imbalance problem in the LV grid. Finally, we address some definitions about hierarchical Petri nets for use in this research.

### 2.1. Urban Microgrid in the Smart-Grid Context

The urban microgrid (UMG) is a special instance of the smart-grid concept, derived from the special architecture of the LV grid inherited from a legacy system existing in several countries [15] and practically in all BRIC (Brazil, Russia, India, China) countries. It imposes that modern forms of power generation in urban unities be integrated with this legacy system to provide a hybrid LV system.

Figure 1 shows a schematic arrangement of the urban microgrid, which is controlled by the MGCC [49]. The UMG derives from a "point of in-common coupling" of the primary grid [18,28,37]. A distributed algorithm is executed by Local Controllers (LC) with a bi-directional communication network [36]. The main goal is to control the energy consumed by domestic loads and integrate the energy flow from distributed energy resources with power converters, and surplus energy into

storage systems. This overall integrated control is managed by a Local Controller Supervisor (LCS), which works as an interface with smart meters [13,19].

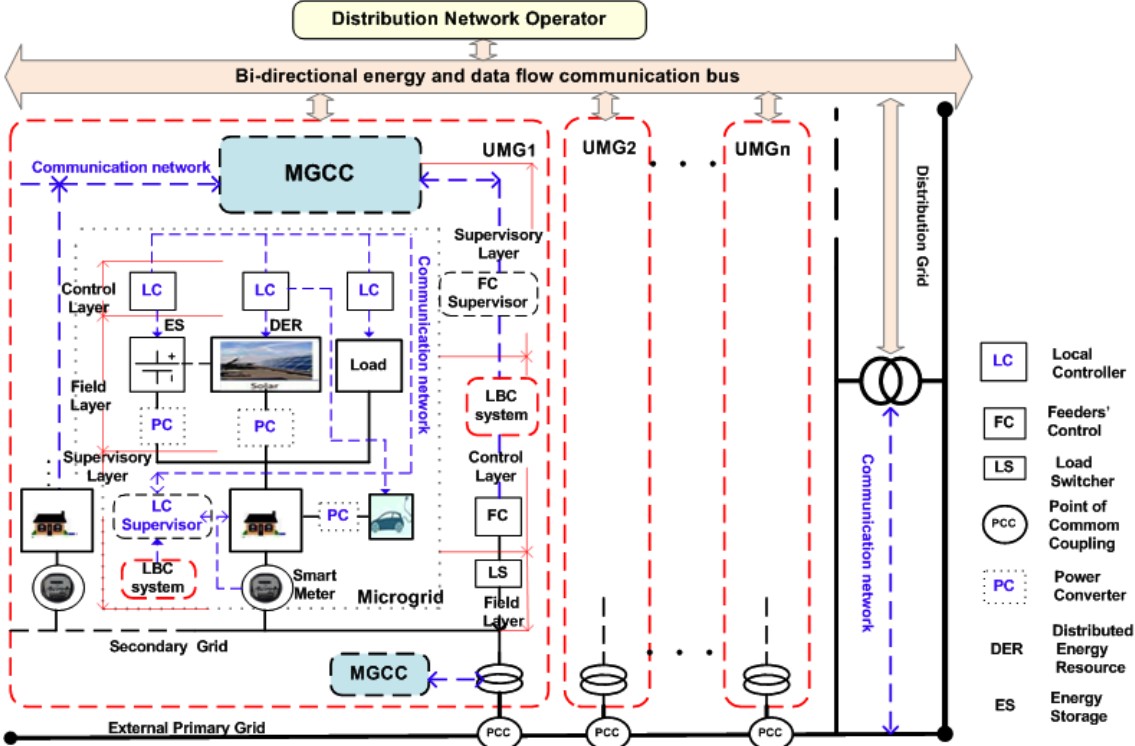

**Figure 1.** General Architecture of Urban Microgrid.

The proposed system is adapted in a global architecture for UMG. Its main component is the LCS, which supervises energy consumption in residential units to identify imbalanced load feeders assuming that a selection can be made for switching in the LV grid. Simultaneously, the Feeder Control Supervisor (FCS) identifies the grid feeder load imbalance, and coordinates the load transfer to reestablish the steady state [38].

### 2.2. Load Imbalance in Low-Voltage Grid Feeders

In the legacy low-voltage grid the "feeder load imbalance" constitutes a power consumption flow problem, as shown in Figure 2.

Generally, it is caused by growing disorder and by unplanned consumption of domestic loads in residences [17]. In extreme situations, this can affect the power supply, especially in the equilibrium between grid feeders. The transformer can be burned if this problem is not solved in good time [50].

The *Phase-Load Balancing* technique based on automatic load switching is an interesting approach for addressing this problem [16], and is an alternative technique to the legacy method used in the most part in LV Brazilian grids [15]. This implies that overloaded single-phase consumer units are switched to a feeder with a lower load level using some electronic switching device, as shown in Figure 2. This uses a control algorithm to automate the load and electrical current minimization [38] or voltage and load [27].

In this paper, instead of load balancing introduced by distributed-resource power injection [10], an automated approach of *phase-load balancing* method will use a control system based on a combined algorithm, addressed in detail in Section 3.3 [51]. In this specific case, we address the control system design using a hierarchical Petri net to achieve an automated and efficient flow for phase-load balancing in the LV grid.

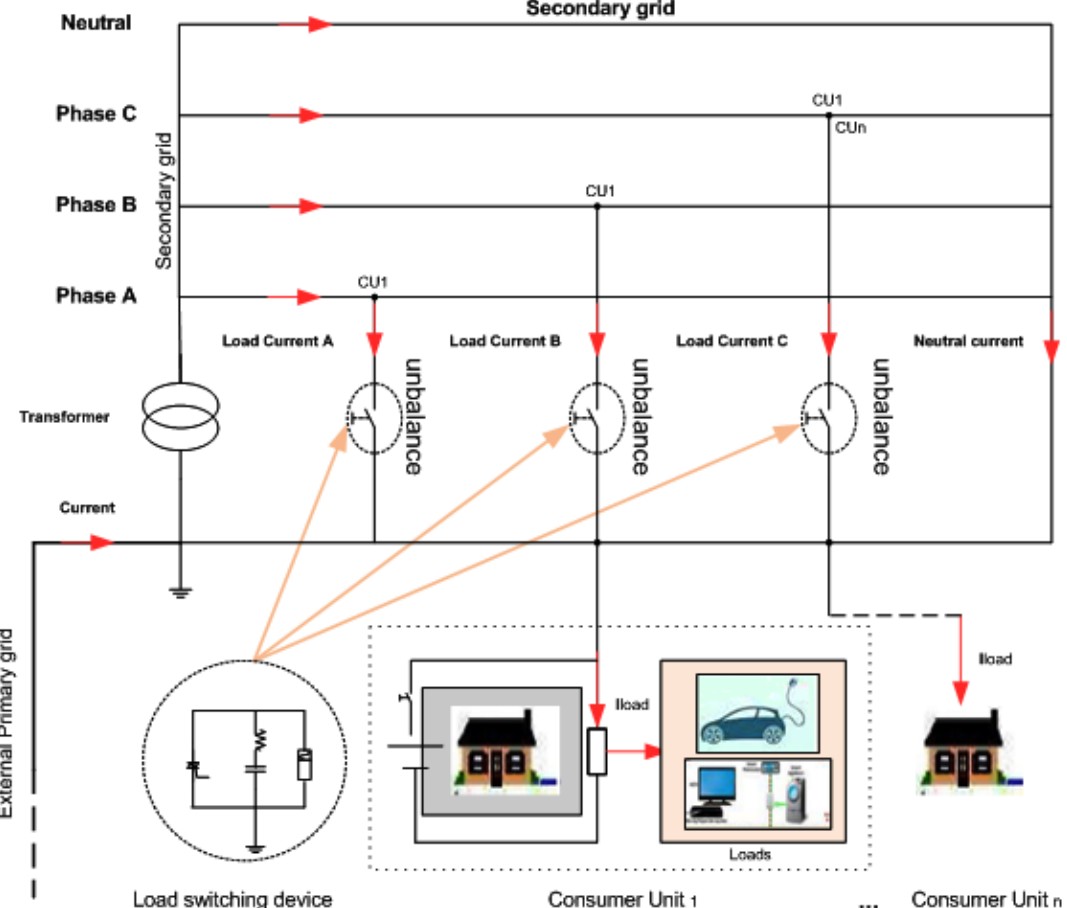

**Figure 2.** Load imbalance in secondary grid.

*2.3. Hierarchical Petri Nets*

Formal models of complex systems—such as urban microgrid control—are validated from static and dynamic points of view by Petri net property analysis and workflow. The use of Petri net extensions could facilitate this process, introducing hierarchical abstraction or time. Among all extensions, we would detach hierarchy and the introduction of time as key issues to support UMG design. The former would be advisable to treat large systems and the latter to open space to optimization in the service provided to user unities (specifically concerning the LBS). Thus, the use of PNs is very suitable since it is a formal method widely adapted to requirements of engineering, due to its wide range of environments for the modeling of dynamics [40,45].

The hierarchical approach would also fit the architecture imposed on the retrofitted (and automated) legacy system [52] and to the identification of points to couple the LBS service. In this article, we will consider a simple case of PNs with four hierarchical levels, which are integrated—the legacy LV grid, the MGCC system, the proposed algorithm system as part of the MGCC, and a fuzzy inference (as part of the proposed system)—for load-balance procedures in LV grid feeders.

2.3.1. Hierarchical Petri Net Definition (HPN)

HPN can be defined as follows.

- *Definition 2.3.1.1. Hierarchical Petri Net (HPN).* A HPN is a 6-tuple, according to expression Equation (1):

$$N = (P, T, A, w, M_0, F) \tag{1}$$

Such that

1.  The 5-tuple

$$B = (P, T, A, w, M_0) \qquad (2)$$

is a marked Petri net, where:

  – $P$ is a finite set of places, $P \neq \varnothing$;
  – $T$ is a finite set of transitions, $T \neq \varnothing$;
  – $A \subseteq (PxT) \cup TxP)$ is the set of arcs from places to transitions and from transitions to places;
  – $w : A \rightarrow \{1, 2, 3, \cdots\}$ is a weight function on the arcs, and
  – $M_0$ is the initial marking of the PN [42]

2.  $F$ is a function *Place-Bounded Substitution* that ensures that a subnet $Y$ limited by transitions can be replaced by a place $s$ generating another net: $N' = \{P', T', F'\}$, where:

  – $P' = P \backslash S_y \cup \{s\}$, where $S_y$ is the set of places in $Y$;
  – $T' = T \cup T_Y$, where $T_Y$ are the transitions in $Y$;
  – $F' = F \backslash Int(Y)$, where $Int(Y)$ is the inner arcs set of $Y$ [46]

In this paper, the use of an HPN is justified by the automated load-balanced flow-integrated system design, where the proposed algorithm system is a subnet of the UMG architecture, which in turn is part also of the LV legacy system. Thus, assessment validation and property verification can be through hierarchy propagation of lower subnets from macro-places using the PBS method.

Thus, through the structure defined in Equation (1), it is possible to model the states and intervals of operations and routines of the workflow, in the form of "P" places, "T" transitions, along with the start and end relationship between each of them in "A" arcs, the sequence order of the workflow in "$M_0$" marking, involving the flows of each level of the integrated distribution system: LV network, microgrid, Load-Balance Control (LBC) system and subsystem of load transfer in "F" hierarchical subnetworks.

In this case, the system design will begin in the legacy LV grid structure, as the first hierarchical level of integrated system, considering the supervision center as the system beginning, i.e., the place and initial marking of the network. The second hierarchical level will be started from the transformers of the LV grid, i.e., the MGCC subnet, in which all the physical structure of automation and control of the load-balancing system will be represented. The third hierarchical level will start from the MGCC control device, i.e., the subnet of the proposed balancing system. In this third subnet, all the programmable steps of the proposed combined algorithm will be represented. Finally, the embedded algorithm subnet used to identify and transfer loads between grid feeders represents the fourth hierarchical level as the formal system design of one of these steps.

## 3. The Load-Balance Control System (LBC)

The proposed system is called load-balancing control (LBC), and is based on a combined algorithm with four stages according to Figure 3, which aims to automate the procedures related to load-imbalance identification in the grid feeders and consumer units, as well as the consumer unit arrangement for the switching process, which is based on load-imbalance level identification and load forecast in the single-phase consumption units [51]. Thus, the system design will be based on this architecture and the LBC system flowchart, as shown in Figure 4.

### 3.1. LBC Architecture

Figure 3 shows the LBC system architecture.

The LBC system interacts with the concessionaire measurement interface, and is composed of:

- Feeder Control Supervisor (FCS). This manages the procedure that identifies the load imbalances in grid feeders, once given from the central control of the legacy LV system. This is based on a fuzzy inference system [38]. Processed load data are collected offline from the MGCC information system. In cases of load imbalance in feeders, it will activate the Local Control Supervisor.

- Local Control Supervisor (LCS). This is activated in cases of imbalance in grid feeders, and performs the load-imbalance identification (based on a fuzzy inference) and load forecasting (based on Markov chains) in single-phase consumption units [51], delivering it as a result in the LC. Data processed as energy, energy variation, and load variation are collected offline from the MGCC information system. Temperature variation and energy price variation are collected offline externally from the meteorology and rnergy market information centers, respectively.

- Local Controller (LC). This receives from the LCS the future states of load consumption and the load-imbalance levels in the single-phase consumers, to chose a switching arrangement. The choice criterion implies selecting consumers that present the highest level of load imbalance and also the highest future state of load consumption in each consumption unit. The choice is checked with the load transfer levels indicated by the FCS in each phase of the grid. The final result obtained the switching arrangement of the consumer units, returning the load stability to the MGCC information system and to the legacy system.

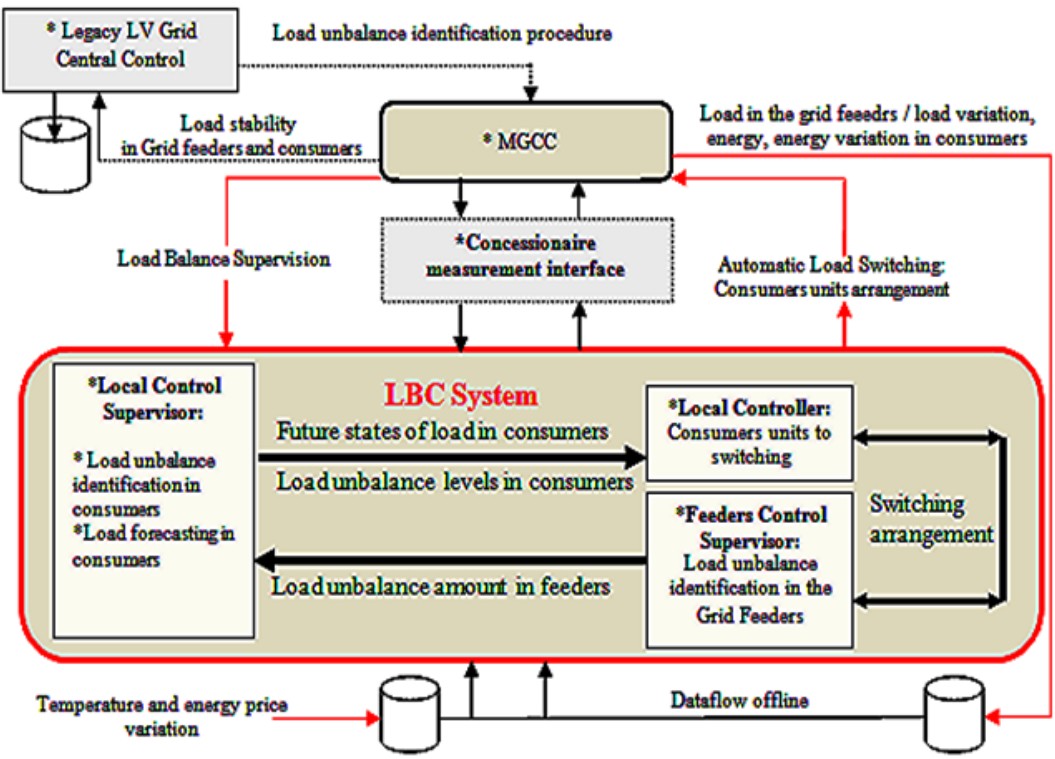

**Figure 3.** LBC Architecture.

### 3.2. High-Level Flowchart of the LBC System

Figure 4 shows in detail the high-level flowchart of the LBC system, as an alternative control to the load-balancing process for the UMG; thus, the LBC system can also be inserted as an interface in the legacy LV grid.

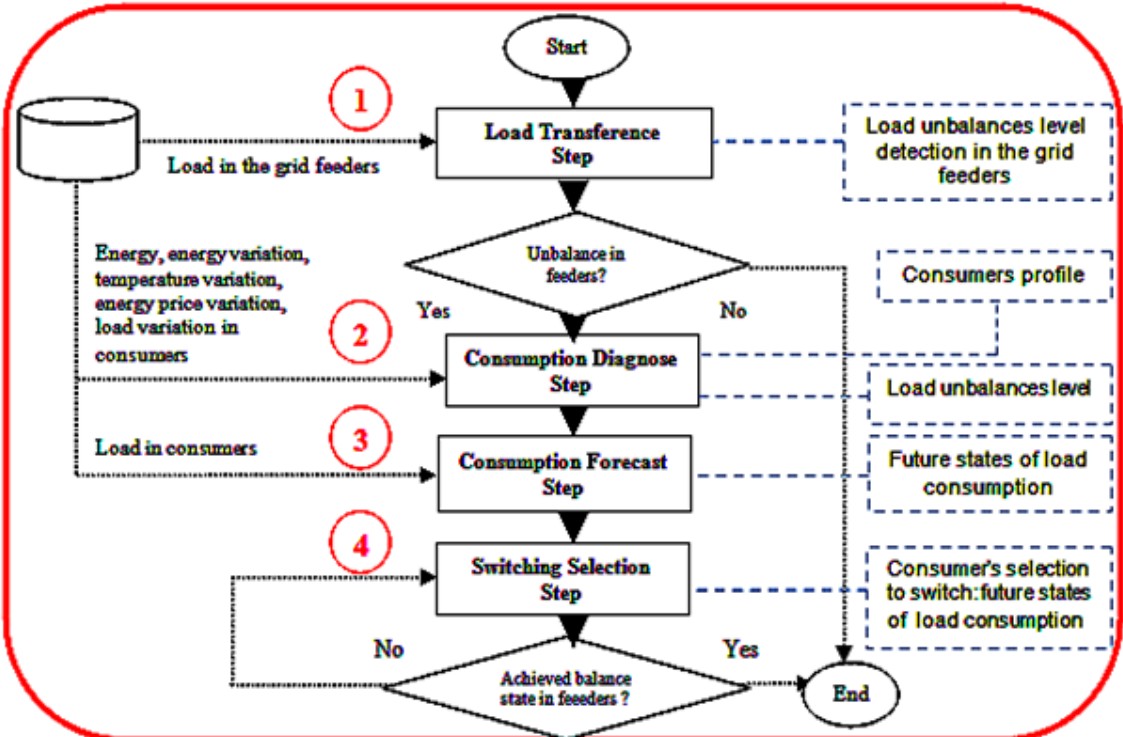

**Figure 4.** LBC system high-level flowchart.

The LBC high-level flowchart is explained as follows:

Step 1. Load Transference. This flow is started when the "Load" level consumptions in each grid feeder (from the database) are processed in the LBC Fuzzy inference (explained in detail in the following subsection) in order to detect load imbalance. As a result, it is informed whether feeders are balanced or not. Thus, both situations are informed by the FCS. In cases of load imbalance in some grid feeders, the second modular step will be started. Otherwise, the process will be ended.

Step 2. Consumption Diagnosis. This module is activated when one of the grid feeders is imbalanced. It is processed in the load-imbalance inference (LUI), also explained in the following subsection, to identify the consumer profile and the load-imbalance level in single-phase consumer units. This result will be used to improve the consumer unit arrangement choice, for the switching process on the grid feeders.

Step 3. Consumption Forecast. This step detects the future load consumption in the single-phase consumption units with load-imbalance levels detected in the previous step. The load future consumption results, along with the load-imbalance level, are used for the consumer unit switching selection on the grid feeders.

Step 4. Switch Selection. This last module assists in obtaining a reliable combination for switching selection of consumer units. It is based on the load future consumption, in each single-phase consumer unit with load-imbalanced level detected. In the case of not finding a good arrangement, a new one will be found, as indicated in the following section. Otherwise, the process will be ended.

*3.3. Combined Algorithms Flowchart of the LBC System*

Figure 5 shows in detail the integration flowchart of the combined algorithms that combines the four programmable steps of the LBC system.

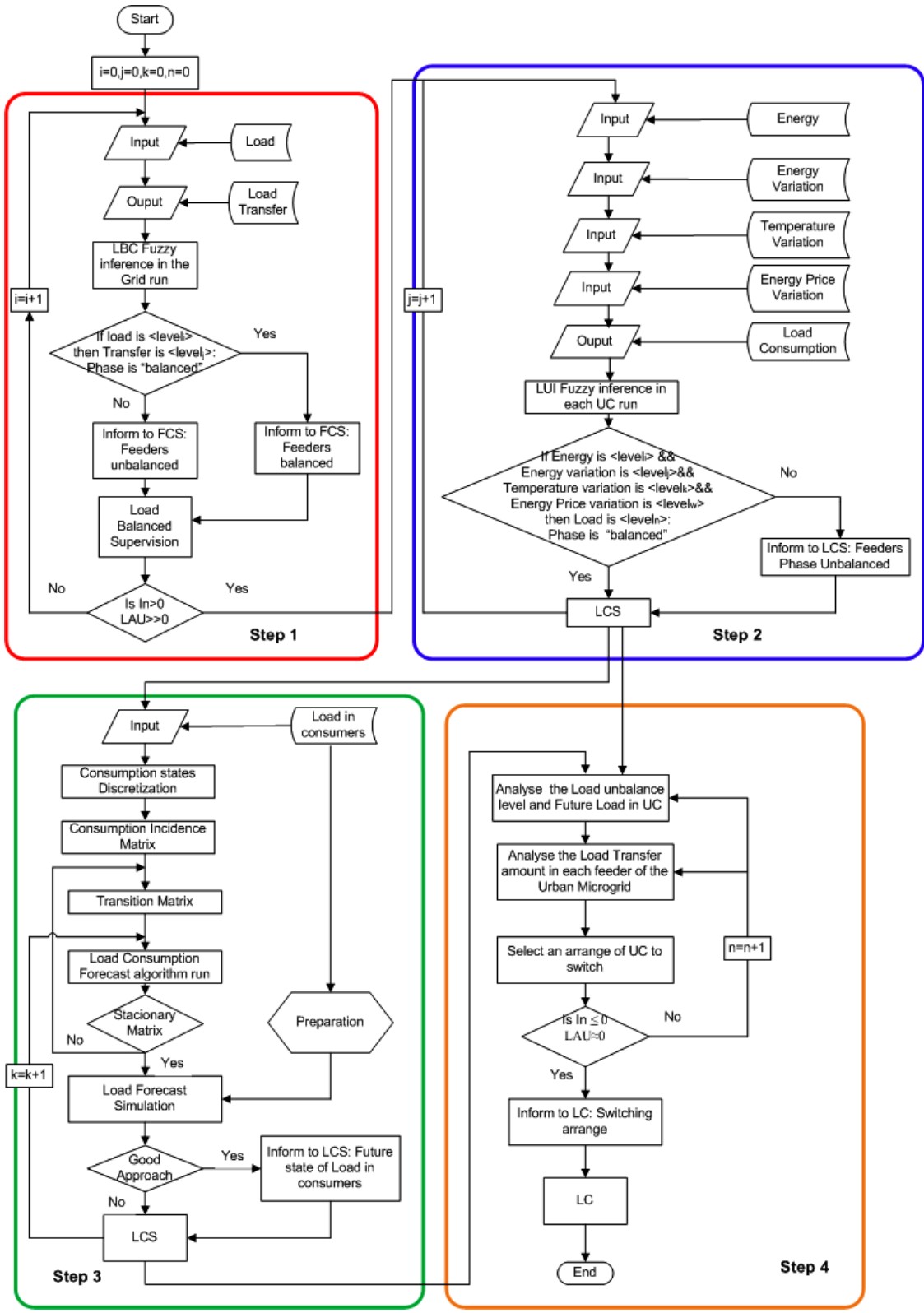

**Figure 5.** Combined algorithms flowchart of the LBC system.

Step 1. **Load Transference inference**. This first algorithm is highlighted in the red rectangle in Figure 5. This is based on a Mamdani's fuzzy inference with only an input called "Load" and an output called "Load Transfer" [51]. The input variable has eight $S_{1_i}$ membership sets, which represent the

possible load level consumptions $x_i$ in each grid feeder with their respective $\mu_i$ membership degree. This is defined according to Equation (3).

$$S_{1i} = \{(x_i, \mu_i(x_i)|x_i \in \text{“Load”}\} \tag{3}$$

where: $i = 1 \ldots 8$.

The output variable has also eight $S_{1j}$ membership subsets, which represent the possible load transference levels $y_j$ to each grid feeder. This is defined according Equation (4).

$$S_{1j} = \{(y_j, \mu_j(y_j)|y_j \in \text{“Load Transfer”}\} \tag{4}$$

where: $j = 1 \ldots 8$.

Thus, both variables are inferred according to Equation (5).

$$\textbf{If } \text{“Load” is “}x_i''\text{” } \textbf{then } \text{“Load Transfer” is “}y_j'' \tag{5}$$

After this process, the FCS is informed that feeders are balanced or lso that feeders are imbalanced. Thus, both situations are informed as to the Load-Balanced Supervision. In cases of load imbalance, the third step will be started. Otherwise, the process will be started again to a new load-imbalance identification procedure.

Step 2. **Load-Imbalance inference**. This second module is highlighted in blue in the Figure 5 and is activated when one of the grid feeders is imbalanced. This is applied only in single-phase consumption units, and this is also based on a Mamdani's fuzzy inference with four inputs called "Energy", "Energy variation", "Temperature variation", and "Energy price variation", and one output called "Load variation" [53]. The input variable definitions are as follows.

- "Energy". This first input variable has three $S_{2_{ai}}$ membership sets, which represent the possible "energy" level consumption $x_{ai}$ in each grid feeder with their respective $\mu_{ai}$ membership degree, according to Equation (6).

$$S_{2_{ai}} = \{(x_{ai}, \mu_{ai}(x_{ai})|x_{ai} \in \text{“Energy”}\} \tag{6}$$

where: $i = 1 \ldots 3$.

- "Energy variation". This second input variable also has three $S_{2_{ai}}$ membership sets, which represent the possible "energy variation" levels $x_{bi}$ in each grid feeder with their respective $\mu_{bi}$ membership degree, according to Equation (7).

$$S_{2_{bi}} = \{(x_{bi}, \mu_{bi}(x_{bi})|x_{bi} \in \text{“Energy variation”}\} \tag{7}$$

where: $i = 1 \ldots 3$.

- "Temperature variation". This third input variable has also three $S_{2_{ci}}$ membership sets, which represent the possible "temperature variation" $x_{ci}$ which affect the consumer units with their respective $\mu_{ci}$ membership degree, according to Equation (8).

$$S_{2_{ci}} = \{(x_{ci}, \mu_{ci}(x_{ci})|x_{ci} \in \text{“Temperature variation”}\} \tag{8}$$

where: $i = 1 \ldots 3$.

- "Energy Price variation". This fourth input variable has also three $S_{2_{di}}$ membership sets, which represent the possible "energy price variation" $x_{di}$ which affect the consumer units with their respective $\mu_{di}$ membership degree, according to Equation (9).

$$S_{2_{di}} = \{(x_{di}, \mu_{di}(x_{di})|x_{di} \in \text{``Energy price variation''}\} \tag{9}$$

where: $i = 1 \ldots 3$.

The output variable has three $S_{2j}$ membership sets, which represent the possible "load variation" $y_j$ in each consumer units with their respective $\mu_j$ membership degree, according to Equation (10).

$$S_{2j} = \{(y_j, \mu_j(y_j)|y_j \in \text{``Current variation''}\} \tag{10}$$

where: $j = 1 \ldots 8$.

Thus, these variables are inferred according to Equation (11).

**If** *"Energy" is "$x_{ai}''$* **and** *"Energy variation" is "$x_{bi}''$* **and** *"Temperature variation" is "$x_{ci}''$* **and** *"Energy price variation" is "$x_{di}''$* **then** *"Load variation" is "$y_{2j}''$* \hfill (11)

In the case of the single-phase consumption units being balanced, the LCS is informed and the process will be started again. Otherwise, the LC is informed of this diagnosis and the process will be started from the third module.

Step 3. **Consumption Forecast**. This third algorithm is highlighted in green in the Figure 5. This step detects the future states of load consumption, for the best choice of the single-phase consumption units to the switching procedure.

This is based on Markov chains performing the load consumption forecast in each "$F_{ij}$" consumer feeder. According to Equation (12), the load data-flow is prepared and is inserted in the input to the Consumption States Discretization. Thus, based on the "$\pi_{ij}$" incidence jump probabilities, the Consumption Incidence Matrix is formed to achieve each "$X(k + n)$" future state (low, medium, and high) from the previous state "$X(k)$". Then, as a result, the Transition Matrix is obtained which starts the Load Consumption Forecast algorithm.

$$C_{F_{\pi_{ij}}}{}^{(n)} = P\{X(k + n) = j|X(k) = i \tag{12}$$

where: $C_{F_{\pi_{ij}}}{}^{(n)} \geq 0$

In cases of not obtaining the Stationary Matrix, the flow will be started again from the Transition Matrix step. Otherwise, the Load Forecast Simulation will be started. In cases of obtaining a good approach, the LCS will be informed of the Future State of Load (FLS) for each consumption unit. Otherwise, the algorithm will run again. The load temporal series validation along a specific period is performed beforehand, training a dataset to establish a reliable forecast model to forecast the FSL. A 48-month data history of load consumption, to forecast 12 months of future consumption, will be used in each consumer in this specific case.

Step 4. **Switch Selection**. This last module is highlighted in orange in the Figure 5. This assists in obtaining an optimal combination to selection of "$i$" single-phase consumer units to switching process, according to Equation (13).

$$L_i = \alpha.min(L_i) + \beta.min(FL_i) \tag{13}$$

Analyzed in Equation (13) is the load variation level detected "$L_i$" and the FLS "$FL_i$" (low, medium, and high), in each consumption unit of the imbalanced feeder, choosing the "$i$" consumer unit that indicates the highest level of "$L_i$" and "$FL_i$". Then, observed in Equation (14) is a restriction of equality, such that the "$L_i$" total load amount of the chosen consumers should not be greater than

the load transfer level "$P_j$" indicated at each "j" phase. In cases of not obtaining a good arrangement, the process will be started. Otherwise, the process will be ended.

$$\sum_{i=1}^{n} L_i \leq P_j \tag{14}$$

## 4. LBC System Implementation Results in Hierarchical Petri Nets

### 4.1. System Design Method

For design purposes, Figure 6 shows the flowchart describing the system design applied method.

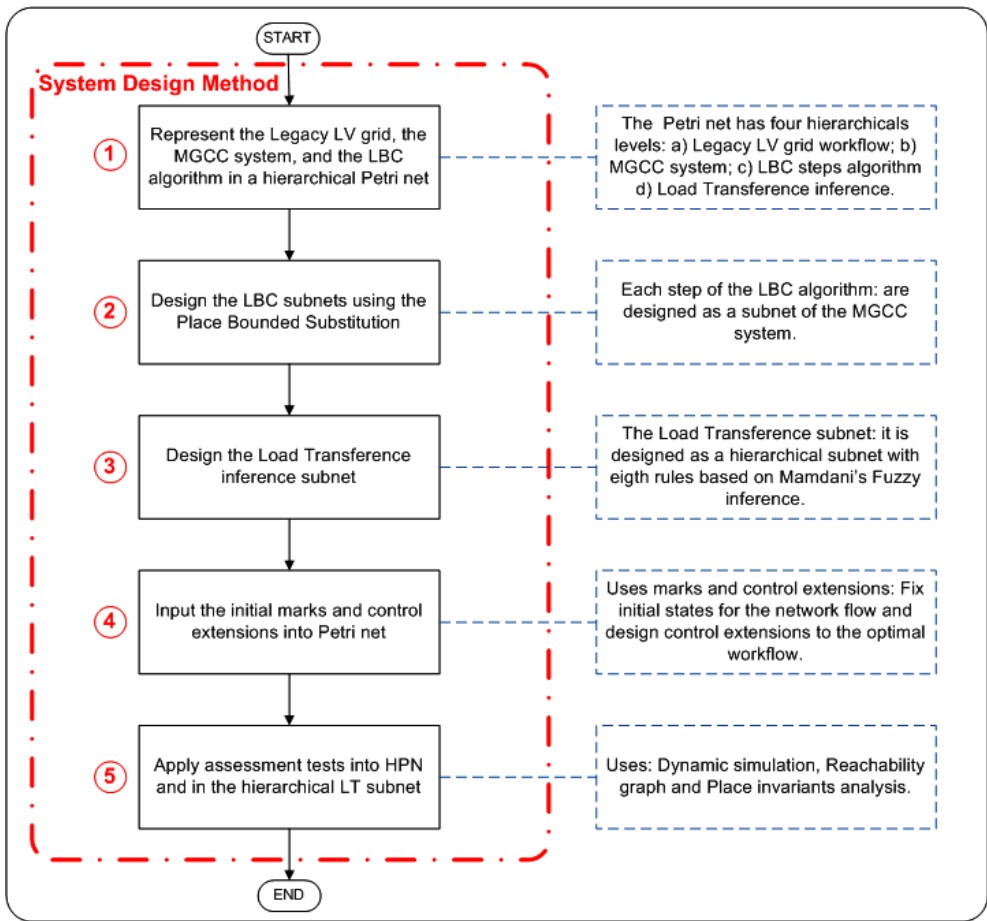

**Figure 6.** Flowchart describing the system design method.

First, system flow integration is represented: the legacy LV grid flow, the MGCC system, the LBC algorithm, and the LT subnet. Each of them composes a level of HPN. In addition, the Load Transference Inference of the LBC system represents a fourth hierarchical subnet, which highlights the formal system design of inference rules for load-balancing procedure in the LV grid. Second, the LBC system design as a subsystem of the MGCC system is performed. Their subnets are represented using thePBS method. Third, the Load Transference Inference system design as a hierarchical LBC subnet is performed, with eight rules (based on Mamdanis' fuzzy inference) to identify load-imbalance level in the grid feeders. Fourth, marks (tokens) on the initial states on network are placed, as well as the control extensions for dynamic simulation. Finally, assessment tests into HPN and the LT subnet are applied. In this case, in both evaluated PNs, the dynamic system simulation, the reachability states analysis to evaluate tangibility of states (places) over HPN and on its subnets, and the place-invariant

analysis to verify compliance of automation routines workflow in the whole HPN and also into its hierarchical subnets will be applied.

### 4.2. Dynamic System Design

Figure 7 shows the LBC system design modeling in an HPN.

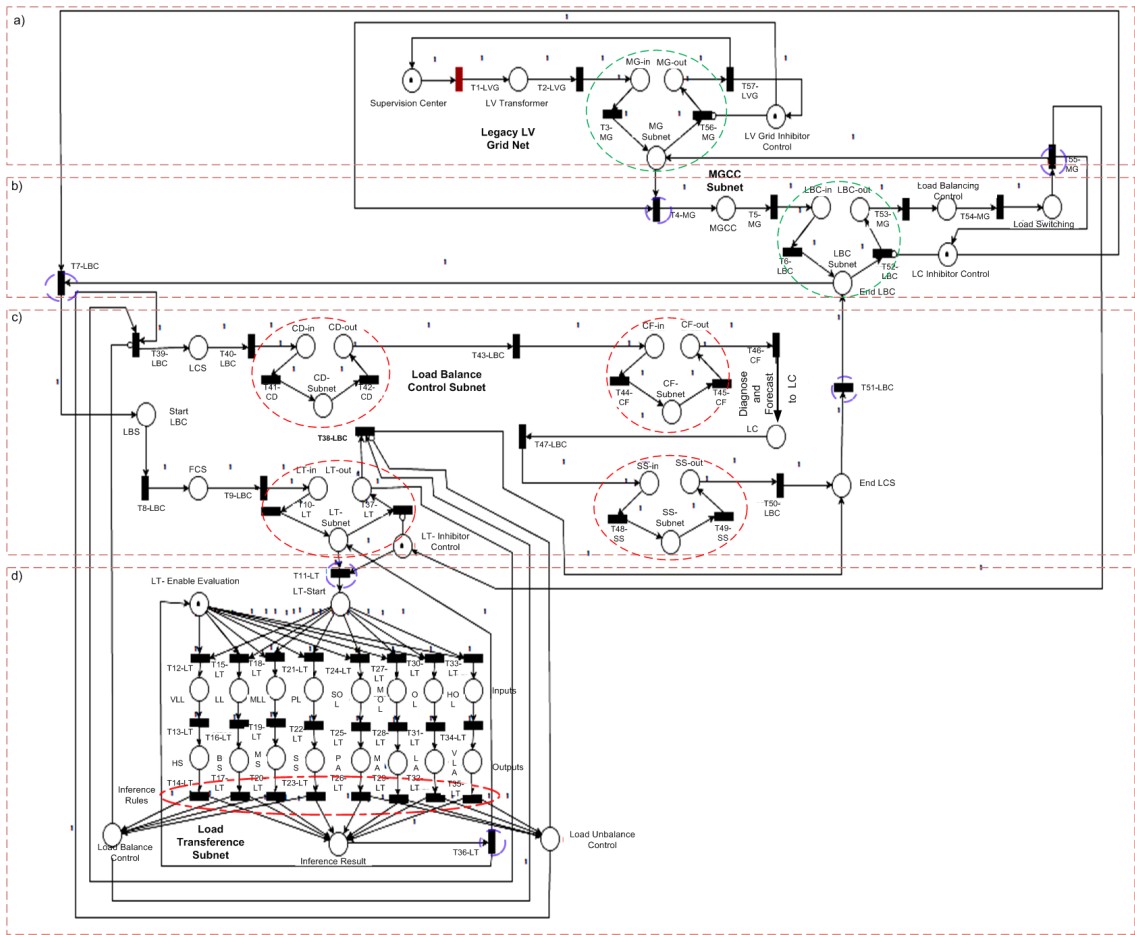

**Figure 7.** HPN System design: (**a**) Legacy LV Grid net; (**b**) MGCC subnet; (**c**) LBC subnet; (**d**) LT subnet.

This model describes four levels of hierarchy:

(a)　**Legacy LV net**. This is shown in Figure 7a on the HPN first level. It represents the currently electrical LV grid, with following operating flow:

　　–　Supervision Center place. This represents the substation supervision center. This starts the whole HPN, and indicates the initial point of load-balancing verification process.

　　–　LV Transformer place. This represents each LV transformer. This has an interface with the load consumption supervision in the secondary grid inner-installed [15]. This also starts control and supervision integration between the new UMG architecture and the MGCC.

　　–　MGCC macro-place. This represents the start of the second HPN level, which is represented by the Place-Bounded Substitution method, highlighted in green circles, with an input place, "$MGU_{in}$" and an output place, "$MG_{out}$" and an "MG Subnet" place. This is bounded by borders formed by the "$T4 - MG$" and "$T55 - MG$" transitions highlighted in blue.

　　–　LV Inhibitor Control. This extension control activates the "$T4 - MG$" transition and inhibits the "$T56 - MG$" transition, ensuring workflow from the first hierarchical level to the second

subnet, avoiding the return to the first subnet without running the full load-balancing flow. It allows only operation flow completion once it is performed, through the "$T55_{MG}$" transition, which, in addition, returns tokens to the LV inhibit control, thus forming an automatic retention property of tokens for this subnet.

(b) **MGCC subnet**. Figure 7b shows this second hierarchical level of HPN, and represents the MGCC architecture addressed by the load-balance control, which follows the operating flow below:

- MGCC place. This represents the MGCC information system [28]. This started the load-balancing procedure in LV grid feeders.
- LBC macro-place. This represents the start of the third HPN level. This is also based on the Place-Bounded Substitution method, highlighted in green circles, with an input place, "$LBC_{in}$" and an output place, "$LBC_{out}$" and the "LBC Subnet" place. This is bounded by borders formed by the "$T7 - LBC$" and "$T51 - LBC$" transitions, highlighted in blue.
- LC Inhibitor Control. This extension control activates the "$T7 - LBC$" transition and inhibits the "$T52 - MG$" transition, ensuring workflow from the second hierarchical level to the third subnet, avoiding return to the second subnet without running full LBC system flow, allowing only flow completion once it is performed, through the "$T51 - LBC$" transition.
- Load-Switching Control place. This represents the load-switching control of consumer units to some LV grid feeders, according to the final result of the LBC system.
- Load-Switching place. This represents the load switching in each LV grid feeder. The final result of the load-balancing process is transferred to the MG subnet through the "$T55_{MG}$" transition. In addition, this transition returns a token to the LC inhibit control, thus also forming an automatic retention property of tokens for this subnet.

(c) **LBC subnet**. Figure 7c shows this third subnet, highlighting in red circles the four LBC flowcharts addressed in Section 3.2 as some specific subnets, derived from a macro-place based on the PBS method. This follows operating flow below:

- LBS place. This represents the Load-Balance Supervision of the LBC system, and the initial state of the third subnet workflow.
- FCS place. This represents the Feeder Control Supervision (FCS) and from it starts the load-balance procedure in each grid feeder.
- LT macro-place. This represents the start of the fourth HPN level. This is also represented by the PBS method, highlighted in red circles, with an input place, "$LT_{in}$" and an output place, "$LT_{out}$" and the "LT Subnet" place. This is bounded by borders formed by input transitions "$T11 - LT$" and by output transition "$T36 - LT$".
- LCS place. This transmits the final detection result of load imbalances from the LT subnet and activates the following subnet.
- Consumption diagnosis (CD) macro-place. This represents the start of "step 2", called the Consumption Diagnosis subnet. This is also represented by the PBS method, highlighted in red circles, with an input place, "$CD_{in}$" and an output place, "$CD_{out}$" and the CD Subnet place.
- Consumption forecast (CF) macro-place. This starts the "step 3", called the Consumption Forecast subnet. This is also represented by the PBS method, highlighted in red circles, with an input place, "$CF_{in}$" and an output place, "$CF_{out}$" and the CF Subnet place.
- LC place. This sends the procedure results from CD and CF subnet to the SS subnet.
- SS macro-place. This represents the start of "step 4", called the Switch Selection subnet. This is also represented by the PBS method, highlighted in red circles, with an input place, "$SS_{in}$" and an output place, "$SS_{out}$" and the SS Subnet place.

&ndash; End LCS place. This represents the final workflow of the LT subnet and transmits as a return to the MGCC subnet.

(d) **LT subnet**. Figure 7d shows the fourth subnet. It represents the load transference based on a Mamdanis' fuzzy machine, which is composed of eight inputs which represent different load levels in each grid feeder. This is shown also in detail in Table 1:

&ndash; VLL place. This represents the heavily less-loaded level in each grid feeder.
&ndash; LL place. This represents the less-loaded level in each grid feeder.
&ndash; MLL place. This represents the medium less-loaded level in each grid feeder.
&ndash; PL place. This represents the perfectly loaded level in each grid feeder.
&ndash; SOL place. This represents the slightly overloaded level in each grid feeder.
&ndash; MOL place. This represents the medium overloaded in each grid feeder.
&ndash; OL place. This represents the overloaded level in each grid feeder.
&ndash; HL place. This represents the heavily overloaded level in each grid feeder.

and by eight outputs that represents different load transfer amounts to each grid feeder. This is shown also in detail in Table 2:

&ndash; HS place. This represents the high subtraction of load level in each grid feeder.
&ndash; BS place. This represents the big subtraction of load level in each grid feeder.
&ndash; MS place. This represents the medium subtraction of load level in each grid feeder.
&ndash; SS place. This represents the slight subtraction of load level in each grid feeder.
&ndash; PA place. This represents the perfect addition of load level in each grid feeder.
&ndash; MA place. This represents the medium addition of load level in each grid feeder.
&ndash; LA place. This represents the large addition of load level in each grid feeder.
&ndash; VLA place. This represents the very large addition of load level in each grid feeder.

Thus, finally, eight associated inference rules are obtained, showed also in detail in Table 3. Each rule is activated one at a time by LT enable evaluation extension control.

&ndash; "$T14 - LT$" transition. This represents the first rule and implies that if load level is "VLL" then "VLA" of load in some grid feeder will be transferred.
&ndash; "$T17 - LT$" transition. This represents the second rule and implies that if load level is "LL" then "LA" of load in some grid feeder will be transferred.
&ndash; "$T20 - LT$" transition. This represents the third rule and implies that if load level is "MLL" then "MA" of load in some grid feeder will be transferred.
&ndash; "$T23 - LT$" transition. This represents the fourth rule and implies that if load level is "PL" then "PA" of load in some grid feeder will be transferred.
&ndash; "$T26 - LT$" transition. This represents the fifth rule and implies that if load level is "SOL" then "SS" of load in some grid feeder will be transferred.
&ndash; "$T29 - LT$" transition. This represents the sixth rule and implies that if load level is "MOL" then "MS" of load in some grid feeder.
&ndash; "$T32 - LT$" transition. This represents the seventh rule and implies that if load level is "OL" then will be transferred "S" of load in some grid feeder.
&ndash; "$T35 - LT$" transition. This represents the eighth rule and implies that if load level is "HOL" then will be transferred "HS" of load in some grid feeder will be transferred.

Thus, as a result of these rules, two possible workflows are addressed:

&ndash; In cases of load imbalance one of last four rules ("$T26 - LT$", "$T29 - LT$", "$T32 - LT$", "$T35 - LT$") will be activated by Load-Imbalance Control following transition "$T39 - LBC$",

activating remaining LBC subnets in sequence until the ending process in the "end LCS"
place.

– Otherwise, in cases of load balance, one of first four rules ("$T14 - LT$", "$T17 - LT$", "$T20 - LT$", "$T23 - LT$" ) will be activated by load-balance control, following transition "$T38 - LBC$" and ending the workflow in the "end LCS" place.

*4.3. System Design Validation*

This section addresses the discussion of dynamic performance of the proposed system implementation. Dynamic simulation assessment, reachability and coverability graph was applied in this case, and the place-invariant analysis was used to verify some properties. In addition, these will also be used to analyze the load transference subnet. A free version of Pipe 4.3.0 was applied a simple case of an extended (hierarchical) play-transition or HPN.

4.3.1. HPN Implementation Analyze

- Dynamic simulation. Figure 7 shows the HPN simulation workflow. Thus, the integrated automation flow between the legacy LV system, the MGCC architecture, the LBC system, and the load transference inference was validated. Thus, it was verified as an extended simple PN with a four-level hierarchy, where each subnet is started from a special macro-place based on the Place-Bounded Substitution method. Through dynamic simulation, all lower hierarchical flows were verified in each subnet and their integration with all upper levels of HPN, complying efficiently the integral workflow addressed in Section 4.2. In addition, several simulations with 10,000 firings were carried out with 50 ms time delay between each firing, and have not been registered as "no stop being" and deadlocks.

- Reachability Graph. Figure 8 shows the reachability graph of the HPN system design.

This represents the PN reachable state diagram obtained from its initial state "$S_0$", indicated by the red arrow. Through this diagram, it was verified that all 52 network states and 57 transitions were reached and covered, without deadlock and conflicts. Thus, Figure 8 shows also two load-imbalance workflow verifications, as a result of the "LT" subnet from the "33" place (highlighted in red circles), called "$LT_{out}$". In this case, both are highlighted in green circles by the "$T38 - LBC$" transition (no load imbalances) and by the "$T39 - LBC$" transition (with load imbalances). In addition, Figure 8 also shows the reachability and coverability of all 19 states of the LT subnet, demarcated from the $T11 - LT$" transition until the $T36 - LT$" transition.

- Place-invariant analysis. Place (P) invariant analysis was performed to verify bounded and liveliness properties of HPN, and especially some automation workflow, which is a set of places marked with the same constant token consumption, ensuring the net completion cycle. In this case, two place-invariant equations were obtained.

Equation (15) shows the first P-invariant that verifies the first automation workflow related to the LT subnet flow: the LT Enable Evaluation starts the LT inference as a control extension, activating one of the possible eight load diagnostic rules to obtain the inference result. This P-invariant shows a lower flow for the load-balancing procedure, and completes marking condition for this cycle equal to "1", while performing an LBC order in the LV grid. Figure 9 shows this workflow highlighted with a red line.

$$
\begin{aligned}
&M(LT - Enable\ Evaluation) + M(VLL) + M(HS) + \\
&M(LL) + M(BS) + M(MLL) + M(MS) + \\
&M(PL) + M(SS) + M(SOL) + M(PA) + \\
&M(MOL) + M(MA) + M(OL) + M(LA) + \\
&M(HOL) + M(VLA) + M(Inference\ Result) = 1
\end{aligned}
\tag{15}
$$

Equation (16) shows the second place-invariant, which verifies the whole flow integration of all hierarchical levels: from the Supervision Center place, the LV transformer place, the MGCC Subnet, and the LBC Subnet, to the LT Subnet to perform the load transference procedure.

$$
\begin{aligned}
&M(Supervision\ Center) + M(LV\ Transformer) + \\
&M(MG_{in}) + M(MG\ Subnet) + M(MGCC) + \\
&M(LBC_{in}) + M(LBC\ Subnet) + M(LBS) + \\
&M(FCS) + M(LT_{in}) + M(LT\ Subnet) + \\
&M(LT - Start) + M(VLL) + M(HS) + \\
&M(LL) + M(BS) + M(MLL) + M(MS) + \\
&M(PL) + M(SS) + M(SOL) + M(PA) + \\
&M(MOL) + M(MA) + M(OL) + M(LA) + \\
&M(HOL) + M(VLA) + M(Inference\ Result) + \\
&M(LT_{out}) + M(LCS) + M(CD_{in}) + \\
&M(CD\ Subnet) + M(CD_{out}) + M(CF_{in}) + \\
&M(CF\ Subnet) + M(CF_{out}) + M(LC) + \\
&M(SS_{in}) + M(SS\ Subnet) + M(SS_{out}) + \\
&M(EndLBS) + M(LBC_{out}) + M(Load\ Balancing\ Control) + \\
&M(Load\ Switching) + M(MG_{out}) = 1
\end{aligned}
\tag{16}
$$

In cases of load imbalances, the CD Subnet, CF Subnet and SS Subnet are activated. The final result is sent for implementation to the load-balance control and the load-switching places. The final report is sent to the MGCC. Thus, a balanced marking for the cycle is equal to "1", and shows that the balanced cycle is a sequential system as already expected. In this case, we also conclude that the combined net including LBC and the coupling with the legacy system is also sequential. Figure 9 shows the balanced workflow highlighted with a blue line.

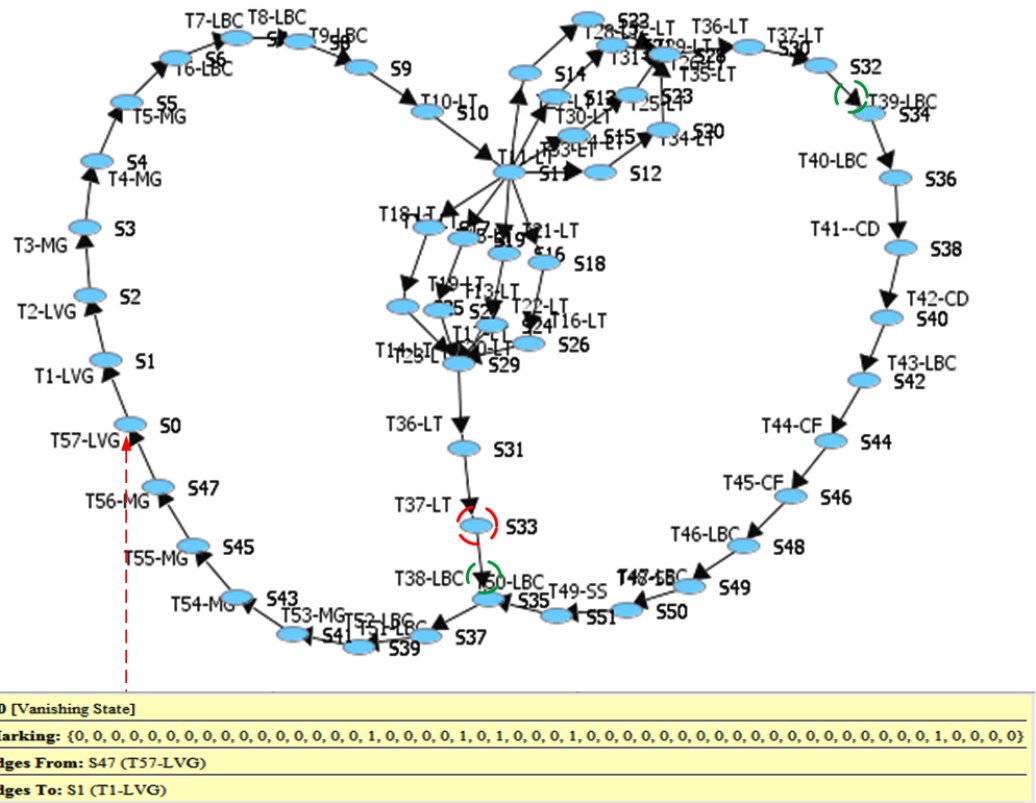

**Figure 8.** HPN Reachability graph.

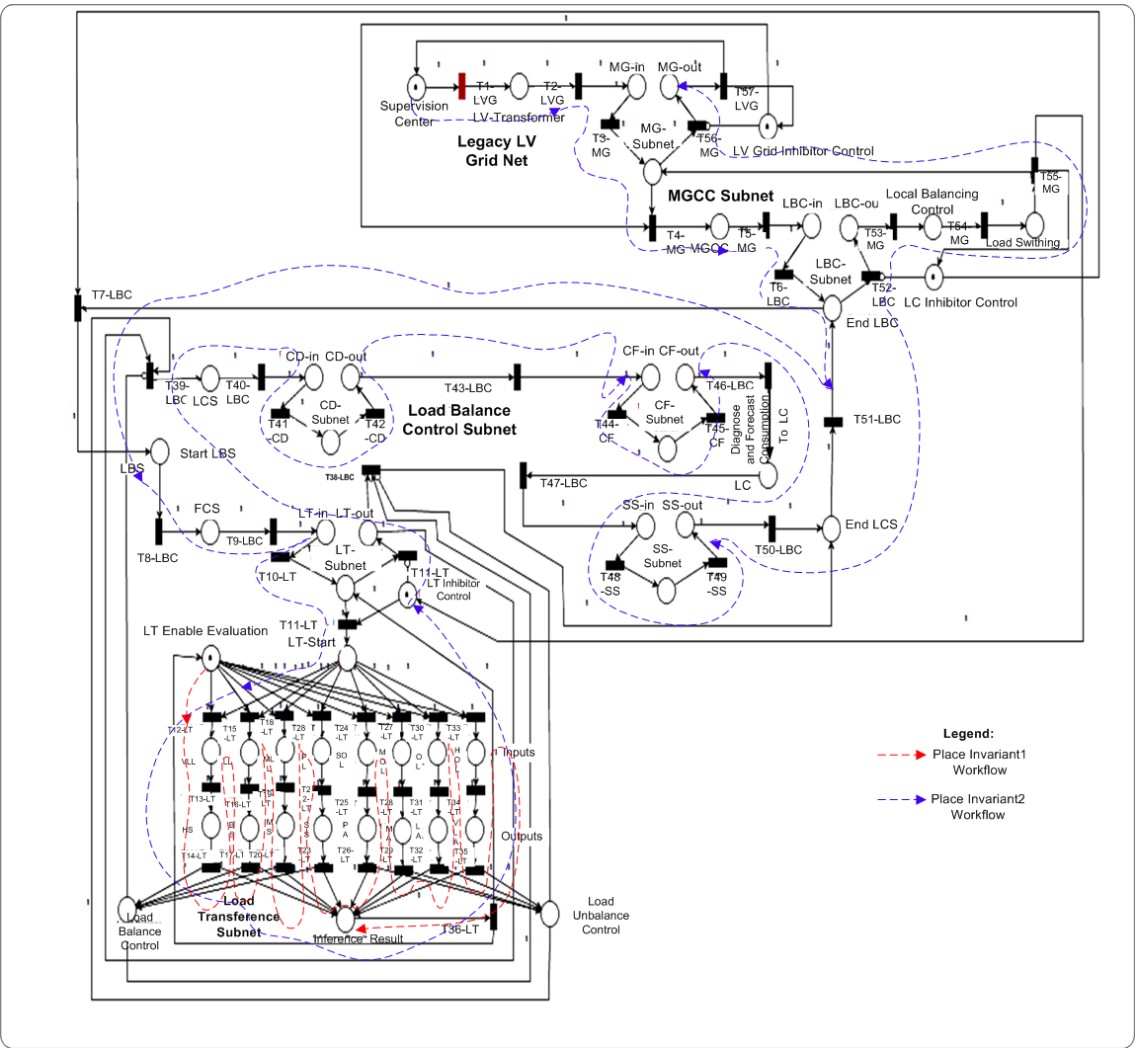

**Figure 9.** HPN Place-invariant workflow representation.

4.3.2. Load Transfer Implementation Analysis

- Dynamic simulation. In this section, the hierarchy is applied in subnets separately to verify the validation and verification into HPN subnets, and to assess the internal automation flow verification with whole network workflows. In this case, Figure 10a shows the LT network system design in detail, with the eight inputs and outputs respectively, as well as, the eight rules inferred highlighted in red.

The pertinence functions of input and output variables are allocated in eight triangular sets, to obtain a homogeneous distribution of the load-imbalance levels in feeders, in the case of the input variable, as well as the transfer levels to load addition or subtraction in feeders, to the output variable. This distribution is reported in Siti [38], where load balancing is applied in a LV circuit, which results in a homogeneous load balancing between feeders, with the lowest load average imbalance level.

Figure 10b shows the membership functions of the input variable parameter "load". Thus, its distribution ranges values are divided into eight sets, and 39.9 Kilo-Watts (KW) was determined as the maximum amount of load allowed in feeders based on the technical data of a 110 kVA transformer with 60 KW of active power [51]. Considering the origin of the first triangular set at 0 KW, the load concentration division was developed manually on the fuzzy toolbox in the MATLAB environment, for each set, and the best obtained distribution is shown in Table 1.

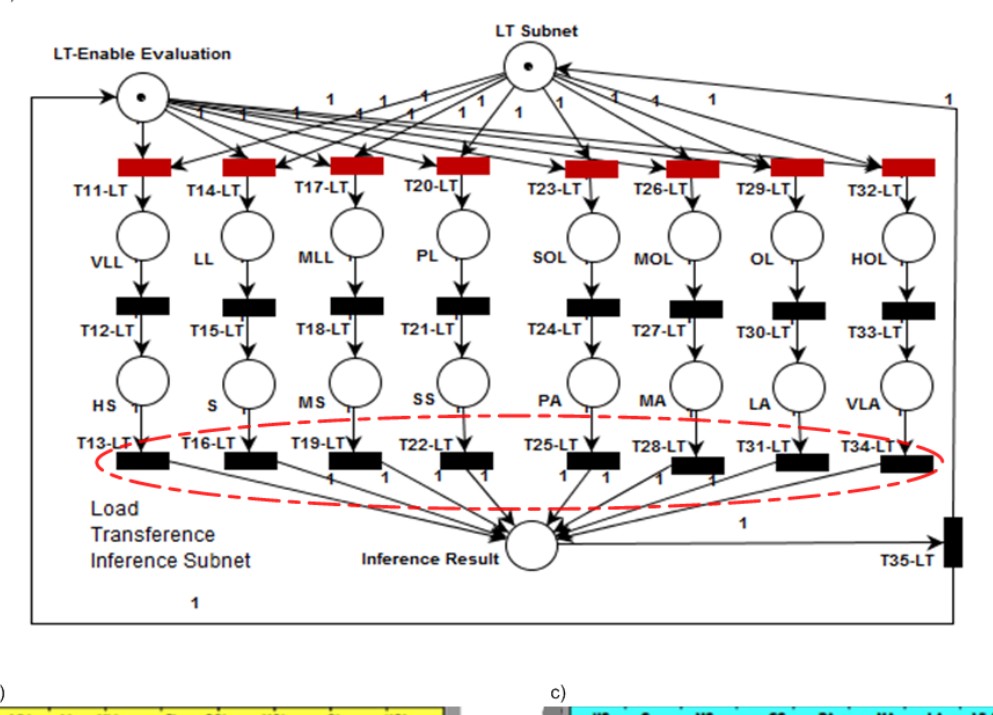

**Figure 10.** Load Transference subnet: (**a**) LT PN; (**b**) Membership function for input parameters; (**c**) Membership function for output parameters.

Figure 10c shows the membership functions of the output variable parameter "load transfer". Similarly, its distribution range values are also divided into eight sets, and "load addition" was considered to the load transfer, in cases of placing additional load in a balance phase, as was "load subtraction" in cases of withdrawing load at an imbalanced phase. −20 KW was determined as the maximum amount of load subtraction, based on the technical data of a LV grid with 110 Kilo-Volts-Amperes (KVA) transformer, and as the maximum amount of load addition, 20 KW [51], as shown in Table 2.

**Table 1.** Input fuzzy nomenclature.

| Inp | Desc | Fuzzy Nom | Kw Range |
|---|---|---|---|
| 1 | Heavily Less-Loaded | VLL | 0–5 |
| 2 | Less-Loaded | LL | 3.8–9.0 |
| 3 | Medium Less-Loaded | MLL | 7.3–13.3 |
| 4 | Perfectly Loaded | PL | 11.8–19.3 |
| 5 | Slightly Overloaded | SOL | 16.3–23.3 |
| 6 | Medium Overloaded | MOL | 21.7–28.4 |
| 7 | Overloaded | OL | 21.2–33.4 |
| 8 | Heavily Overloaded | HL | 32.3–39.8 |

Table 3 shows the *fuzzy* rules for the LT system. Thus, it was verified as an simple PN subnet with 19 places and 25 transitions. This is started in the LT Subnet place as an extended hierarchical level of the LT macro-place of HPN. Through dynamic simulation, all inference evaluation flows were verified,

and the inference results were also verified as transitions highlighted in red circles. In addition, several simulations with 10,000 firings were carried out with 50 ms delay between each firing, and have not been registered as "no stop being" and deadlocks.

**Table 2.** Output fuzzy nomenclature.

| Out | Desc | Fuzzy Nom | Kw Range |
|-----|------|-----------|----------|
| 1 | High subtraction | HS | −20 to −15.3 |
| 2 | Big subtraction | BS | −16.5 to −10 |
| 3 | Medium subtraction | MS | −12.9 to −3.6 |
| 4 | Slight subtraction | SS | −4.9 to −2 |
| 5 | Perfect Addition | PA | 0–6 |
| 6 | Medium Addition | MA | 5.0–11.2 |
| 7 | Large Addition | LA | 10.1–15.7 |
| 8 | Very large addition | VLA | 15–20 |

**Table 3.** Fuzzy rules.

| Rule | If Input | Is | Then Output | Is |
|------|----------|-----|-------------|-----|
| 1 | "Load" | VLL | "Transfer" | VLA |
| 2 | "Load" | LL | "Transfer" | LA |
| 3 | "Load" | MLL | "Transfer" | MA |
| 4 | "Load" | PL | "Transfer" | PA |
| 5 | "Load" | SOL | "Transfer" | SS |
| 6 | "Load" | MOL | "Transfer" | MS |
| 7 | "Load" | OL | "Transfer" | BS |
| 8 | "Load" | HOL | "Transfer" | HS |

- Reachability graph. Figure 11 shows the reachability graph of the LT subnet. It represents the PN reachable diagram obtained from its initial state "$S_0$" highlighted by a red circle, which also represents the initial marking of this PN. Through Figure 11 it is verified, the reachability and coverability of all 19 states and 25 transitions of the LT subnet were reached and covered without deadlock and conflicts. Thus, being verified also each possible rule inferred in order to each specific level of load concentration (input variables) and load transference amount (output variables) of the "Fuzzy" system design, addressed in Section 4.2.

On the other hand, it is observed that this result coincides with the reachability and coverability states diagram found in the HPN net to the LT subnet, shown in detail in Figure 8, due to the hierarchy propagation to the lower subnets, thus verifying the tangibility of the evaluation of inputs, outputs, and rules of the load-balancing inference system to grid feeders.

- Place-invariant analysis. Equation (17) shows the first P-invariant that verifies the first automation workflow of the LT Subnet.

$$
\begin{aligned}
&M(LT - Enable\ Evaluation) + M(VLL) + M(HS) + \\
&M(LL) + M(BS) + M(MLL) + M(MS) + \\
&M(PL) + M(SS) + M(SOL) + M(PA) + \\
&M(MOL) + M(MA) + M(OL) + M(LA) + \\
&M(HOL) + M(VLA) + M(Inference\ Result) = 1
\end{aligned}
\tag{17}
$$

In this case, evaluation of each inference rule based on the inputs and outputs variables following the stream is verified: "LT Enable evaluation", which evaluates "VLL place" and "HS place" to perform the first inference rule, "LL place" and "BS place" to perform the second inference rule, "MLL place" and "MS place" to perform the third inference rule, "PL place" and "SS place" to perform the fourth inference rule, "SOL place" and "PA place" to perform the fifth inference rule, "MOL place" and "MA

place" to perform the sixth inference rule, "OL place" and "LA place" to perform the seventh inference rule and "HOL place" and "VLA" to perform the eighth inference rule.

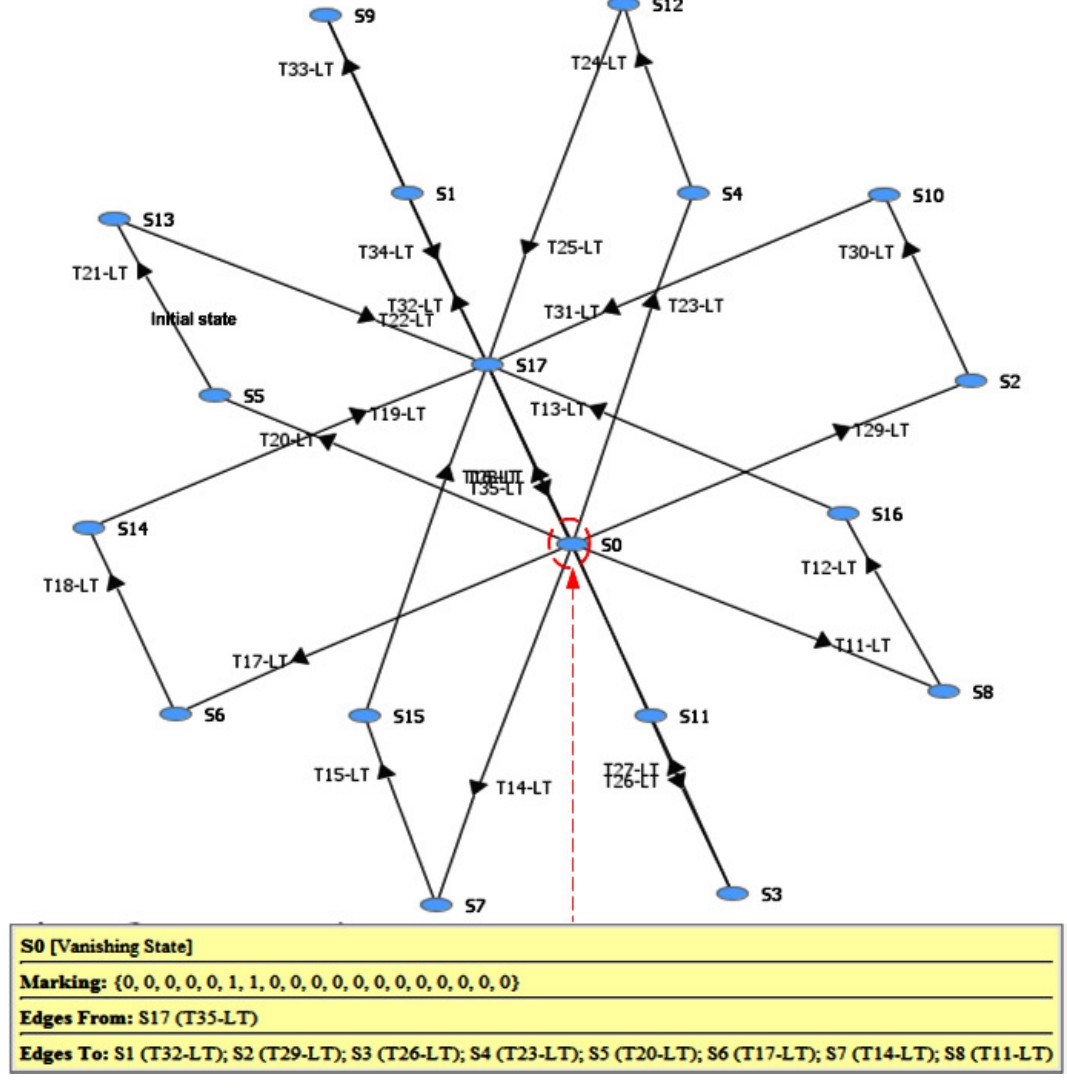

**Figure 11.** Reachability graph of the Load Transference subnet.

The final result is sent to the inference result. Thus, the complete marking condition of this sequence of places is be equal to "1", while performing this evaluation process.

In addition, the internal flow of load-imbalance identification, through propagation of hierarchy of the "LT macro-place" to the Load Transference Subnet is also verified in this subnet. Figure 12 shows this workflow highlighted with a red line.

By contrast, Equation (18) shows the second P-invariant, which verifies the workflow of the Load Transference Subnet: from input variables are started from the "LT Subnet place": "VLL place", "LL place", "MLL place", "PL place", "SOL place", "MOL place", "OL place", "HOL place", and started also the outputs variables: "HS place", "BS place", "MS place", "SS place", "PA place", "MA place", "LA place", and "VLA place".

$$
\begin{aligned}
&M(LT\,Subnet) + M(VLL) + M(HS)+\\
&M(LL) + M(BS) + M(MLL) + M(MS)+\\
&M(PL) + M(SS) + M(SOL) + M(PA)+\\
&M(MOL) + M(MA) + M(OL) + M(LA)+\\
&M(HOL) + M(VLA) + M(Inference\,Result) = 1
\end{aligned}
\tag{18}
$$

Thus, these are associated with the inference rules that are transferred as a result of the "inference result place". This procedure is performed through the internal control "LT Enable Evaluation" place, which enables only a rule selection after the evaluation of each one, thus, emulating a fuzzy Mamdani inference, evaluating each condition (set) of the membership function of the input variable "Load", and each condition (set) of the relevance function of the output variable, "Load Transfer".

This guarantees the best choice evaluation for load transfer, according to the situation identified in each phase, regardless of load subtraction, when the phase is imbalanced, or load addition when there is a balanced phase.

Thus, this validates the cycle and completes marking condition of the sequence be equal to "1", the final result back to the "LT Subnet place" from where it will be propagated to the upper-hierarchical levels of HPN by the "LT macro-place". Figure 12 shows this place-invariant flow, highlighted with a blue line.

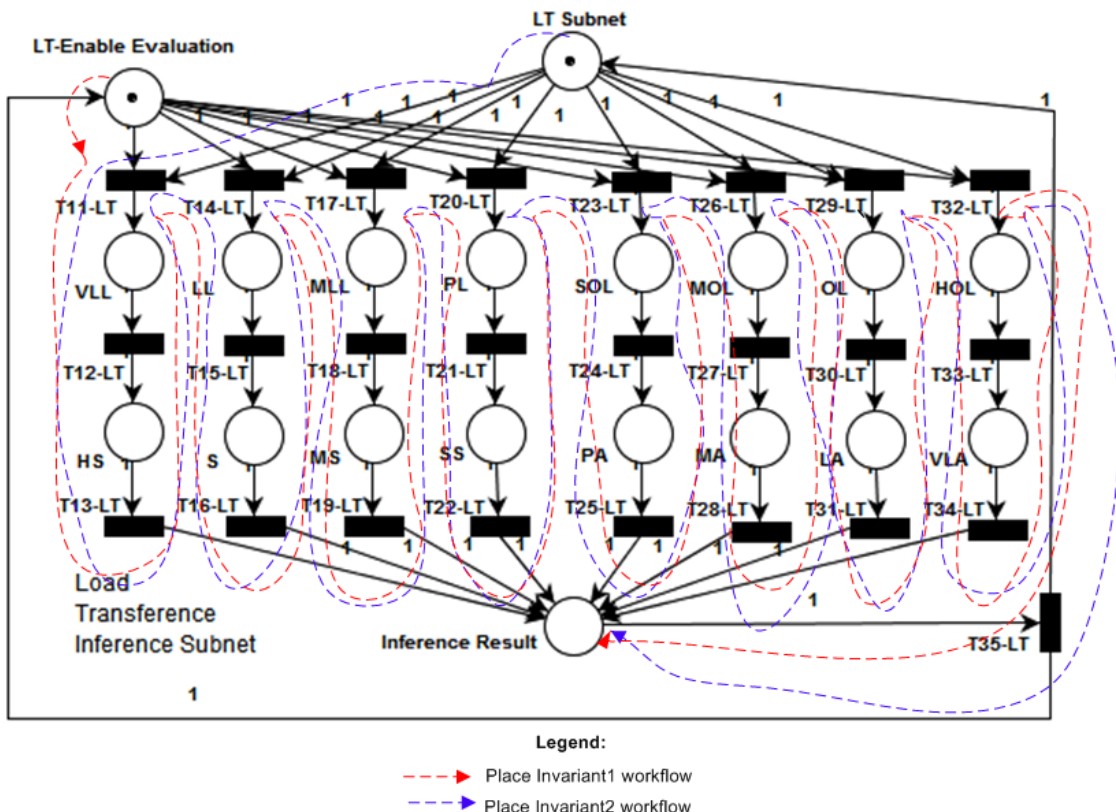

**Figure 12.** LT subnet- Place-Invariant workflow representation.

Thus, the automation flow verification of the integrated system (HPN) and the load transfer algorithm (LT subnet) are validated by the evaluation of the two place invariants obtained in each case, respectively. The existence of these place invariants demonstrate, first, that the workflow of the load transfer system works inviolably, reliably and efficiently, without risks of stoppages or infinite cycles, and that the workflow of the integrated system also works inviolably, reliably and efficiently without risk of stoppages or infinite cycles.

### 4.4. LBC HPN Performance Results Evaluation

Based on the results obtained through the dynamic simulation, the reachability and coverability graph and the place-invariant analyses, it is possible to perform the following dynamic and performance evaluation of the integrated network (HPN) and the LT subnet.

Through dynamic simulation, it was possible to observe that the integrated workflow of the main integrated network (HPN) and the load transference algorithm (LT subnet) reached all its states and transitions, without identification of conflicts, bottlenecks, siphons, and temporary shutdowns or deadlock. This guarantees an efficient and faultless automatic flow of the algorithm proposed in all its internal stages, and especially in relation to its integrated automation flow with the upper-hierarchical levels—in this case, the MGCC system and the legacy LV grid.

The obtained results show that the attainability and coverage graph was generated, which ensures that the HPN and the LT subnet growth are limited, respectively, without the incidence of infinite cycles of routines (which may be caused by trapped siphons and deadlock), therefore making both networks limited. On the other hand, the results also ensure that all states and transitions are reached, in the integrated workflow and in each upper-hierarchical level, thus verifying the hierarchy propagation in both networks.

As a result of the place-invariant analysis, it is first verified that the internal workflow of the LT subnet is guaranteed inviolably without stops, conflicts, or deadlock, due to the constant marking consumption of the places set which constitute it, as indicated in Equation (15) and (17), therefore ensuring, in each case, the only admissible flow for the load transfer process at the grid feeders.

It is also verified that the integrated workflow for the HPN was guaranteed in an inviolable way without stops, conflicts, or deadlock, with the incidence of constant consumption of mark in the places that constitute it, as indicated in Equations (16) and (18). Through this the invariant workflow between the proposed balancing algorithm and the integrated system was checked, as well as the hierarchy propagation in each layer of the network and of integral flow, thereby ensuring the only admissible flow for the integrated automation process between the LBC system and the upper-hierarchical supervision and control levels.

Therefore, based on the above-mentioned analyses, a load-balancing algorithm in the LV grid was obtained, with reliable, efficient, and secure flow, without conflicts, stops, or deadlock, acting in an efficient and secure manner with the internal steps (algorithms and subroutines) and with the upper control and supervision systems of the MGCC and the legacy BT network.

*4.5. LBC Simulation Results and Test Performance Evaluation*

To validate the performing of the obtained system design, the LBC system was submitted to a simulation study with real data (referring to load consumption in September of 2015), in a LV circuit of Manaus city (a north Brazilian city) with load consumption data of 51 consumers, a transformer of 110 KVA, with almost 67 KW of active power.

The load distribution in each grid feeder is broken down as shown in Table 4, and it is verified that there is a phase-load imbalance, as indicated in Equation (19) that shows an initial absolute balance (IAB) level per phase of 13.33 KW.

$$\frac{IAB}{phase} = \frac{(|F_A - F_B| + |F_B - F_C| + |F_C - F_A|)}{3} = 13.33 \ (\text{KW}) \tag{19}$$

The neutral current is $I_N$, determined by the currents in each phase, according to Equation (20).

$$I_N = I_{F_A} + I_{F_B} + I_{F_C} = 38.28(A) \tag{20}$$

As a next step, we will show the results obtained from the application of the LBC algorithm applied in the load-balancing process of the circuit shown in Table 4, identifying the load amount to be subtracted in the imbalanced phases and added in the balanced phases. The concentration levels and the future states of the load consumption states are then considered for the choice of single-phase consumers for switching process in the grid feeders, according to the workflow of the LBC algorithm validated in Section 4.3.

Thus, the results of each step of the LBC algorithm are shown, considering for this purpose its application in the LV grid feeder under study, and the performance evaluation of the CD step and the CF step of one of the single-phase consumer units (CU) of the phase A, with 0.5 KW (360 KWh) highlighted in bold in Table 4.

Table 5 shows the load transfer for each grid feeder: 20 KW to phase A, 21 KW to phase B and 20 KW to phase C, in comparison with the original imbalanced. Subtract 12 KW from phase A, and add 4 KW in phase B and 8 KW in phase C, respectively.

**Table 4.** Load Consumption Data in a LV grid.

| CU-$P_A$ | KW | CU-$P_B$ | KW | CU-$P_C$ | KW |
|---|---|---|---|---|---|
| 1 | 2.0 | 21 | 0.6 | 21 | 0.5 |
| 2 | 2.3 | 22 | 0.1 | 37 | 0.1 |
| 3 | 1.6 | 23 | 0.6 | 38 | 1.3 |
| 4 | 1.2 | 24 | 1.0 | 24 | 0.8 |
| 5 | 1.0 | 5 | 0.6 | 39 | 0.2 |
| 6 | 1.8 | 25 | 0.1 | 40 | 0.6 |
| 7 | 1.8 | 26 | 0.1 | 41 | 0.1 |
| 8 | 1.5 | 27 | 1.5 | 42 | 0.1 |
| 9 | 0.7 | 9 | 0.5 | 9 | 0.2 |
| 10 | 2.5 | 28 | 1.7 | 43 | 1.8 |
| 11 | 2.0 | 29 | 1.0 | 44 | 0.1 |
| 12 | 0.2 | 30 | 1.2 | 45 | 0.6 |
| 13 | 1.8 | 31 | 1.5 | 46 | 0.1 |
| 14 | 2.5 | 32 | 0.1 | 47 | 1.6 |
| 15 | 2.4 | 33 | 0.1 | 48 | 0.5 |
| 16 | 2.7 | 34 | 0.1 | 49 | 1.4 |
| 17 | 1.0 | 17 | 1.0 | 17 | 0.5 |
| 18 | **0.5** | 35 | 1.7 | 50 | 0.2 |
| 19 | 1.5 | 36 | 2.5 | 51 | 1.0 |
| 20 | 1.,0 | 20 | 1.0 | 20 | 0.3 |
| $P_A$ | 32 | $P_B$ | 17 | $P_C$ | 12 |

**Table 5.** LT-step results.

| Scenary | $I_N$ (A) | Phase A (KW) | Phase B (KW) | Phase C (KW) | LAU (KW) |
|---|---|---|---|---|---|
| Imbalanced | 38.28 | 32 | 17 | 12 | 13.3 |
| LBC | 0 | 20 | 21 | 20 | 0.6 |
| Load Transfer | | 12 | 4 | 8 | |

Thus, the procedure for the efficient load transfer between phases implied in verifying the diagnosis and prediction of the future states of load consumption, in the single-phase consuming units of the phases where the loads were subtracted, in this case phase A. Table 5 shows the load transfer for each grid feeder: 20 KW to phase A, 21 KW to phase B and 20 KW to phase C, in comparison with the original imbalance. Subtract 12 KW from phase A and add 4 KW in phase B and 8 KW in phase C, respectively, eliminating the neutral current and significantly attenuating the average imbalanced load, around 0.6 KW.

The results of the CD step are shown in Table 6, and indicate the load limits allowed in each phase for three discrete levels of consumption (low, medium, and high), depending on the energy variation (EV), the temperature variation (TV), and energy price variation (EP) as addressed in Section 3.3. On the other hand, the CF step results indicate the monthly consumption forecast with twelve steps forward, i.e., the future value of load (FL) for three states of consumption (low, medium, and high). Based on these two results, a future consumption matrix for 12 months of 2015 was implemented and is shown in Table 7, where the first column indicates the discrete load consumption states projected for

each month. The asterisk values (of the consumption states) are the values, where the forecast was not correct.

In other columns are the EV, TV, EP variation (PV) and load variation (LV).

As indicated in the last column, the diagnosis for switching selection can be "To switch" (S) in case of load variation indicating a value greater than 0.3. Otherwise, "Do not switch" (NS).

**Table 6.** Load level limits in single-phase.

| Load Level Consumption | Load Variation (%) |
|---|---|
| Low | >0.2 |
| Medium | >0.3 |
| High | >0.4 |

**Table 7.** Future Consumption Matrix of single-phase CU.

| Month | CF Step | EV | TV | PV | LV | Diag |
|---|---|---|---|---|---|---|
| J | Low | 0 | 0, 1 | 0.2 | <0.3 | NS |
| F | Medium | 0 | 0.1 | 0.1 | <0.3 | NS |
| M | Medium | 0.3 | 0.1 | 0.2 | <0.3 | NS |
| A | Medium | 0.3 | 0.2 | 0.1 | <0.3 | NS |
| M | Medium | 0.5 | 0.2 | 0.1 | <0.3 | NS |
| J | Medium | 0.1 | 0.3 | 0.1 | <0.3 | NS |
| J | Medium | 0.2 | 0.3 | 0.2 | <0.3 | NS |
| A | Medium | 0.35 | 0.4 | 0.3 | >0.3 | S |
| S | High | 0.36 | 0.4 | 0.3 | >0.3 | S |
| O | High | 0.37 | 0.4 | 0.3 | >0.3 | S |
| N | Medium | 0.2 | 0.3 | 0.3 | <0.3 | NS |
| D | Medium | 0.3 | 0.2 | 0.3 | <0.3 | NS |

Table 7 shows the results applied in the single-phase CU "18" of phase A with 0.5 KW, with 360 KWh of energy consumption for the month of September. From a history of consumption of 48 months, the discrete consumption states of low consumption (100 KWh), medium consumption (165 KWh), and high consumption (240 KWh) are distributed, obtaining the future consumption projections for each month of 2015 according to the second column of Table 7, through the algorithm indicated in Equation (12).

It shows the future consumption matrix for this consumer unit, specifying in the month of September (study analysis period) to switch (S) because the load variation in the phase is greater than 0.3, due to "High" value of FL in this month, applying the same procedure to the other single-phase CU of this phase, as shown in Table 8.

**Table 8.** Diagnosis Matrix for Load Transfer of Phase A.

| CU | Diag | Load (KW) | CU | Diag | Load (KW) |
|---|---|---|---|---|---|
| 1 | S | 2.0 | 11 | S | 2.0 |
| 2 | S | 2.3 | 12 | S | 0.2 |
| 3 | NS | 1.6 | 13 | S | 1.8 |
| 4 | S | 1.2 | 14 | S | 2.5 |
| 6 | NS | 1.8 | 15 | NS | 2.4 |
| 7 | NS | 1.8 | 16 | S | 2.7 |
| 8 | S | 1.5 | 18 | S | 0.5 |
| 10 | S | 2.5 | 19 | S | 1.5 |

In this case, the single-phase CU, 2, 10, 14, 16, 18, and 19 were subtracted from phase A, totaling 12 KW. In phase B, the CU 10 and 19 were added, totaling 4 KW, and in phase C, the CU 2, 14, 16 and 18 were added, totaling 8 KW. This results in a final load-balance state as shown in Table 5.

To compare the results, we consider the legacy load-balancing method, a method based on a fuzzy balancing algorithm [38], a LBC algorithm approach (LBC1), and a LBC2 approach, considering an optimal solution to load imbalance. All performances of each applied method were developed using the MATLAB environment. These results are showed in Table 9.

**Table 9.** Load-Balance Performance.

| Param | Imbalance | Legacy | Fuzzy | LBC1 | LBC2 |
|---|---|---|---|---|---|
| $L_{P_A}$ KW | 32 | 25 | 22 | 20 | 19.8 |
| $L_{P_B}$ KW | 17 | 12 | 19 | 21 | 20 |
| $L_{P_C}$ KW | 12 | 24 | 20 | 20 | 20 |
| $I_N$ A | 38.3 | 0 | 0 | 0 | 0 |
| IAB KW | 13.3 | — | — | — | — |
| LAU KW | — | 8.7 | 2.0 | 0.6 | 0.1 |

Figure 13a shows the load transfer in each phase, according to each of the methods applied, the load distribution represented in green, obtained by the proposed system (LBC1).

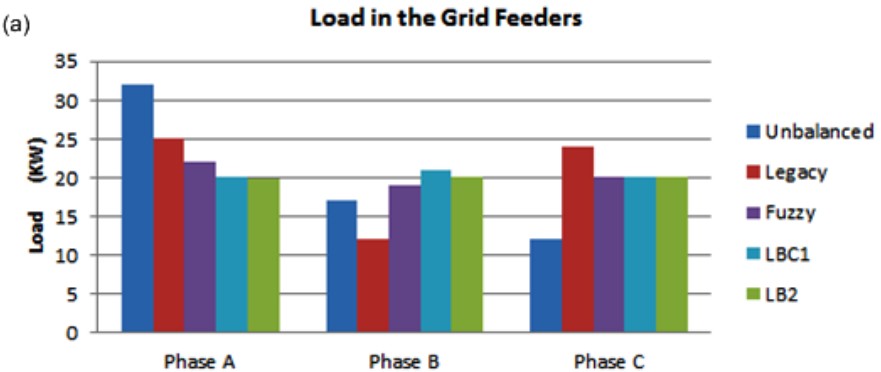

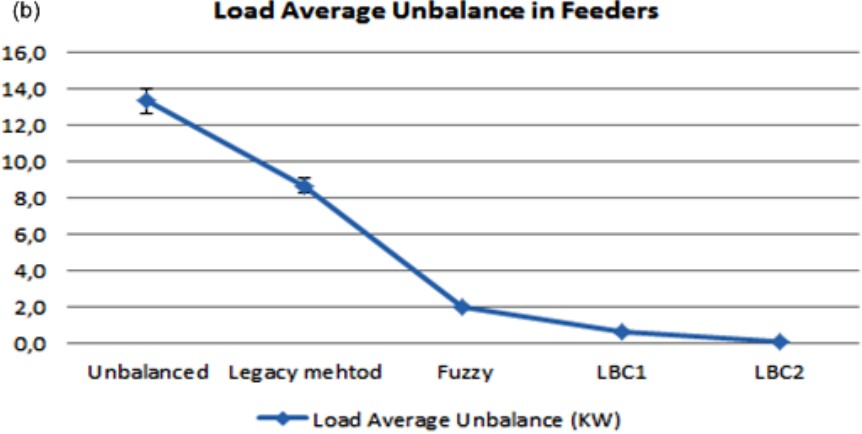

**Figure 13.** LBC system validation: (**a**) Load in the Grid Feeders; (**b**) Load absolute imbalance.

In addition, the LBC system (LBC1) reached the lowest mean load-imbalance value, around 0.6, compared to the legacy system method results of around 8.7 KW , and a fuzzy control algorithm, with around 2 KW, therefore proving its efficient validation which is showed also in Figure 13b. Finally, through a second system application (LBC2) a lowest load average imbalance (LAU) value around 0.1 KW was obtained, indicating a load division of 19.8 KW in phase A and 20 KW in phases B and C respectively. However, this solution is applicable when switching from the consumer unit "4" with 1.2 KW to another LV circuit.

Based on results obtained, the efficiency of the proposed method is demonstrated, in relation to the intelligent identification of load transfer in each phase of the LV circuits, and the choice of the CU for the switching process, according to their level of concentration and future states of the load consumption, thus ensuring an effective and reliable balancing process, as well as the load balancing in feeders.

### 4.6. Future Practical Implementation in the Control System of UMGs

The validated model becomes an alternative control proposal for load balancing in the legacy LV grid where the implementation of microgrids and distributed sources of power generation is not currently implemented. Thus, it can act as an alternative control resource in synchronization with the current injection of microgrids, the integrated coordination of control and the integrated coordination of multimicrogrids, as discussed in Section 1 and the general architecture of urban microgrids, as also discussed in Section 2.1 and according to the bibliographic review [21,31,32]. Where it would constitute a distributed control system connected with the CU, the LV transformer, the MGCC system, and the supervision center of the whole electrical system, in order to guarantee the efficient acquisition of consumption data, the load-balancing application and the switching selection according to the load consumption matrix of each single-phase consumer unit.

By contrast, the implementation of the LBC system as a combined system of integrated algorithms will mainly obey the workflows validated in this work, through specific semantics of structural language translation of embedded systems still in development by the authors. Each step of the combined algorithm will have operational modularity synchronized with all stages of the system. Its experimental validation will be carried out first in circuits of the LV legacy grid, and later in urban microgrids, in order to validate its effectiveness as an alternative control for the balancing of LV grids.

### 5. Conclusions

In this paper, a new system design of a distributed control system for load-balance procedure in the LV grid has been presented. This is composed of combined algorithms, called LBC system, contributing to the load amount identification to transfer between feeders, and, with the single-phase consumer unit selection, to the switch operation of load-balance procedure. In this case, a hierarchical PN approach was used first to represent and to validate the workflow of each inner algorithm of the control system. In this case, the inner algorithm of load transfer identification (LT subnet) was developed to highlight the fuzzy inference employed in the intelligent identification of the amount of load to be withdrawn or added in LV grid feeders. In addition, we represent and also validate the integrated workflow of the proposed system with the upper-hierarchical levels, as the MGCC system, and the legacy LV grid. The PBS method was used to represent the hierarchical-level connection of network. This was developed using macro-places formed by an input and output, as well as the initial location of the lower subnet.

Both networks were tested through dynamic simulation, the application of the reachability and coverability graph, and the place-invariant analysis. Verifying reliable and reliable dynamic performance in both, free of conflicts, stops and deadlock, the attainability of all its states and transitions was also verified, identifying that both are limited and safe networks. Finally, two inviolable workflows were identified in both networks, which guarantee the efficient execution of the load transfer algorithm and its evaluation of each fuzzy inference rule used to identify load transfer, respectively, as well as the integrated workflow between the LBC system with upper-hierarchical control and supervision levels in the MGCC and the LV grid. This provided an efficient and reliable load-balancing algorithm that ensures a single and admissible load-balancing (automation cycle) solution to the integrated control workflow, as well as a unique and admissible inference rule to the load transfer.

The combined algorithm of the LBC system was also tested by dynamic simulation above the historical load of a LV circuit with a transformer of 110 KVA, which presented load imbalance between its phases.

The results showed the identification of the load transfer amount in each phase, as well as the limits of variation of load in relation to the discrete states of consumption in each phase, the future consumption matrix that indicates the switching diagnosis of each single-phase consumer unit in relation to their limits of load variation, and the future load consumption states. The consumer unit selection was based on the diagnosis of this matrix for the month of September 2015. The performance of the LBC system (LBC1) was compared along with the legacy load-balancing method, a fuzzy controller, in relation to the load transfer in each phase, and the load average imbalance (LAU) value. The LBC system presented the lowest LAU, around 0.6 KW, compared to the other applied methods. A second application of the LBC system (LBC2) was also tested, presenting the lowest mean load-imbalance value, around 0.1 KW, demonstrating the efficiency of the proposed system.

For future work, the authors propose the development of a coordinated control system to represent the electrical current injection from microgrids and the LBC system as a simultaneous, integrated and automated operation flowchart, in order to efficiently ensure the load management consumption and greater load stability in UMG. This new model will be developed using timed transitions with fixed intervals of operation, to emulate the workflow temporal integration along the integrated UMG and each lower hierarchical layer.

**Author Contributions:** Conceptualization, J.R.S. (Jose R. Sicchar); Data curation, J.R.S. (Jose R. Sicchar); Formal analysis, J.R.S. (Jose R. Sicchar), J.R.S. (Jose R. Silva) and C.T.D.C.J.; Investigation, J.R.S. (Jose R. Sicchar); Methodology, J.R.S. (Jose R. Sicchar), J.R.S. (Jose R. Silva), R.C.O. and W.D.O.; Supervision, C.T.D.C.J. and J.R.S. (Jose R. Silva); Validation, J.R.S. (Jose R. Sicchar); Data Analyzed, J.R.S. (Jose R. Sicchar), J.R.S. (Jose R. Silva) and C.T.D.C.J.; Visualization, J.R.S. (Jose R. Silva), R.C.O. and W.D.O.; Writing—original draft, J.R.S. (Jose R. Sicchar), J.R.S. (Jose R. Silva), C.T.D.C.J., R.C.O. and W.D.O.

**Acknowledgments:** The authors thank to UEA, UFPA, USP and FAPEAM, for allowing scientific achievement of this proposal.

**Conflicts of Interest:** The authors declare no conflict of interest.

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
