# Peer review of "A Load-Balance System Design of Microgrid Cluster Based on Hierarchical Petri Nets"

_energies, doi:10.3390/en11123245_

Reviewer 1 Report

Comments to the authors

Manuscript Number: energies-382302

Title: Load Balance System Design to Microgrid cluster based on Hierarchical Petri Nets

1) Literature review of the paper can be enhanced by considering more recent and relevant works. For example:

1-1) “Fault-tolerant supervisory controller for a hybrid ac/dc micro-grid”, IEEE Transactions on Smart Grid, 2018.

1-2) “A Comprehensive Strategy for Accurate Reactive Power Distribution, Stability Improvement, and Harmonic Suppression of Multi-Inverter-Based Micro-Grid”, Energies, 2018.

1-3) “Adaptive Virtual Impedance Droop Control Based on Consensus Control of Reactive Current”, Energies, 2018.

1-4) “Robust Optimal Power Management System for a Hybrid AC/DC Micro-Grid”, IEEE Transactions on Sustainable Energy, 2015.

1-5) “Control and EMS of a Grid-Connected Microgrid with Economical Analysis”, Energies, 2018.

2) In this paper triangular membership functions are used to built the fuzzy controller. Please justify the values (Tables 1 and 2) and shape of the membership functions.

3) How can one guarantee that the flowchart given in Figure 5 does not get stuck, i.e., it provides an admissible and unique solution.

4) The bar graphs in Figure (13) are not clear. Please try to use different colors to make it more understandable.

Author Response

The authors would like to thank the anonymous reviewers for their very constructive comments and suggestions to help us improve the quality of our manuscript. 

Reviewer #1:  1) Literature review of the paper can be enhanced by considering more recent and relevant works.

Author’s response: Firstly the authors thank the careful revision of the reviewer. The bibliographical references were studied and also inserted in the context of the introduction of the article, is indicated in the last paragraph of page 1 and in the first three paragraphs of page 2 of the article. 

Reviewer #1: In this paper triangular membership functions are used to built the fuzzy controller. Please justify the values (Tables 1 and 2) and shape of the membership functions.

Author’s response:  The pertinence functions of input and output variables are allocated in eight triangular sets, in order to obtain a homogeneous distribution of the "load unbalance levels" in feeders, in case of the input variable, as well as the "transfer levels" to load addition or subtraction in feeders, to the output variable. This distribution is reported in Siti \cite{siti-2011distribution}, where load balancing is applied in a LV- circuit, which results in a homogeneous load balancing between feeders, with the lowest load average unbalance level.

On another hand, the values of Table 1 ( ranges of the input variable, called Load) are justified as follow: the  distribution ranges values are divided into eight sets, and it was determined as the maximum amount of load allowed in feeders, 39.9 kW, based on the technical data of a 110 kVA transformer with 60 kW of active power follow the similar method applied also in Sicchar \cite{sicchar-controlador}.

Considering the origin of the first triangular set at “0” kW. The load concentration division was developed manually on the Fuzzy toolbox of Matlab environment.

Similarly, the values of Table 2 (ranges of the output variable, called “Load Transfer”) are justified as follow: the distribution ranges values are divided into also eight sets, and it was considered to the load transfer, "load addition" in case of placing additional load in a balance phase, and "load subtraction" in case of withdrawing load at an unbalanced phase. It was determined as the maximum amount of load subtraction, -20 kW, based also on the technical data of a LV-grid with 110 KVA transformer, and as the maximum amount of load addition, 20 KW  follow also the method reported in Sicchar \cite{sicchar-controlador}.

Reviewer #1: How can one guarantee that the flowchart given in Figure 5 does not get stuck, i.e., it provides an admissible and unique solution.

Author’s response: The obtaining of a single and admissible solution for the load balancing is guaranteed through the analysis of place invariants applied in the model developed in Petri nets for the LBC system (HPN). It shows a sequence of inviolable sequence flow with constant consumption of marks from a single set of places following the operations of the LBC algorithm and its integration with the upper hierarchical control layers of the MGCC system and the central supervision of the low voltage network, and a second flow for the choice of only one rule of inference for the transfer of charge. This is reinforced by the two place-invariants of the load-transfer (LT) subnet, with the first flow being the general flow of the load-transfer algorithm, the choice of only one rule after the evaluation of the eight rules by the Fuzzy controller.

Reviewer #1: The bar graphs in Figure (13) are not clear. Please try to use different colors to make it more understandable.

Author’s response: Figure 13 was redone with colors that highlight in the first illustration 13-(a), to the result of the transfer of load in each phase obtained by the applied methods, the legacy system, a Fuzzy controller of the bibliographic revision, the system LBC (LBC1) highlighted in green, and a second application of the LBC system (LBC2). The second illustration in figure 13-(b) also shows the performance of the LBC system and the other methods applied, in relation to the load average unbalance (LAU), where the LBC system also obtained the lowest value.

Reviewer 2 Report

It is important to discuss load balance in quantitative manner, Fig 13 is not adequate.

mapping qualitative and quantitative will be useful to add.

review English.

Explain FL with quantitative values.

discuss practical implementation and how to link to existing energy systems in real world

Author Response

Reviewer #2:  1) It is important to discuss load balance in quantitative manner, Fig 13 is not adequate, mapping qualitative and quantitative will be useful to add.

Author’s response: Figure 13 was redone with colors that highlight in the first illustration 13-(a), to the result of the transfer of load in each phase obtained by the applied methods, the legacy system, a Fuzzy controller of the bibliographic revision, the system LBC (LBC1) highlighted in green, and a second application of the LBC system (LBC2). The second illustration in figure 13-(b) also shows the performance of the LBC system and the other methods applied, in relation to the load average unbalance (LAU), where the LBC system also obtained the lowest value. 

Reviewer #1: Explain FL with quantitative values.

Author’s response:  Table 7 shows the results applied in the single-phase CU "18" of phase-A with 0.5 kW, with 360 kWh of energy consumption for the month of September. From a history of consumption of 48 months, the discrete consumption states of low consumption (100 kWh), medium consumption (165 kWh) and high consumption (240 kWh) are distributed. Obtaining the future consumption projections for each month of the year 2015 according to de second column of Table 7, through the algorithm indicated in equation 12.

Reviewer #1: discuss practical implementation and how to link to existing energy systems in real world.

Author’s response:  The validated model becomes an alternative control proposal for load balancing in the legacy LV- grid where the implementation of microgrids and distributed sources of power generation is not currently implemented. Thus, it can act as an alternative control resource in synchronization with the current injection of microgrids, the integrated coordination of control and the integrated coordination of multi-microgrids, as discussed in section \ ref{Int} and the general architecture of urban microgrids as also discussed in subsection \ ref{urban} and according to the bibliographic review \cite{dong2018comprehensive}, \cite{lyu2018adaptive}, \cite{el2018control}. Where it would constitute a distributed control system connected with the consumer units, the LV-transformer, the MGCC system, and the supervision center of the whole electrical system, in order to guarantee the efficient acquisition of consumption data, the load balancing application and the switching selection according to the load consumption matrix of each single-phase consumer unit.

On the other hand, the implementation of the LBC system as a combined system of integrated algorithms will mainly obey the workflows validated in this work, through a specific semantics of structural language translation of embedded systems still in development by the authors. Each step of the combined algorithm will have operational modularity synchronized with all stages of the system. Its experimental validation will be carried out first in circuits of the LV- legacy grid, and later in urban microgrids, in order to validate its effectiveness as an alternative control for the balancing of LV-grids.

Reviewer 3 Report

1. In Section 1, the authors firstly explained the works relating to the problems and so on. This section is not well written, and it did not address the major issue that is imperative to be solved. As a reviewer’s point of view, the literature survey of this section is very weak, unfocused and insufficient. What is the essential problem of this work? The authors should really explain the drawback of approaches in related works especially instead of simply stating what they have done. The authors should discuss the mentioned references in the introduction part.

2. What is the major novel/contribution of this paper? In my view, there are many techniques adopted in the recent past for this problem. So, the authors should improve the section with the references. Please explain the main contribution related to previous approaches, and provide a list of paper’s contributions at the end of the introduction.

3. The abstract is not properly representative of the entire paper. Please make sure that your abstract is properly structured. In some places, the text should be revised to be more clear.

4. In Section 6, the authors are encouraged to provide a greater depth of discussion (more details), and modify the discussion and conclusion as well.  

5. Figs 1, 2, 3, 4, 5, 6, 7, 8, 9, 10, 13 are not clear. They are very difficult to read. Please rewrite all these figures. Unless the authors can clearly conduct, this paper should not be accepted.  

Author Response

Reviewer #3:  1) In Section 1, the authors firstly explained the works relating to the problems and so on. This section is not well written, and it did not address the major issue that is imperative to be solved. As a reviewer’s point of view, the literature survey of this section is very weak, unfocused and insufficient. What is the essential problem of this work? The authors should really explain the drawback of approaches in related works especially instead of simply stating what they have done. The authors should discuss the mentioned references in the introduction part.

Author’s response: Section 1 of the article was rewritten indicating the correct bibliographical revision of the contextualization of the problem, and of the proposal. Starting  with a review of techniques used for load balancing in the low-voltage grid feeders, in the context of intelligent urban microgrids. Addressing characteristic of each of them, and their disadvantages. Directing the discussion to the use of the phase load balancing method, due to its application in developing countries with little presence of microgrids implementation and distributed generation. This is followed by the contribution of the Petri nets to the wide range of property verification features, to the problem of the modeling gap and formal verification of intelligent control algorithms and supervision of load balancing in the low voltage grid.

Reviewer #3: What is the major novel/contribution of this paper? In my view, there are many techniques adopted in the recent past for this problem. So, the authors should improve the section with the references. Please explain the main contribution related to previous approaches, and provide a list of paper’s contributions at the end of the introduction.

Author’s response:  The authors believe that the main contribution of this paper is on providing a formal process automation model that optimizes and integrates the workflow of a load balancing control system in the legacy LV-grid. The proposed control system is based on combined algorithms to minimize load consumption in the grid phases (feeders), through following programmable procedures: "load transfer in the grid feeders", which is based on a Fuzzy inference to identify and perform the load transfer or between feeders; "unbalances consumers units identification", which is based on a Fuzzy inference system to detect the load unbalances level in the consumers units; "load forecast in consumers units", which is based on Markov chain algorithm that forecast the monthly levels of discrete states on load consumption; and "switch selection" which is based on an optimal choice algorithm of unbalanced consumers units with high load consumption.

The main contribution of this paper can be summarized as follow:

-It is proposed a novel system design of a load balance system integrated with the legacy LV-system and urban microgrids. This is validated in Petri nets, emphasizing the novel form of encapsulating combined algorithms, evidenced by hierarchy levels of integration; \cite{wang-2015knowledge};

- Use of the reachability graph and place-invariants analysis for properties verification and the experimental assessment of robustness and efficiency of the load balance algorithm. In addition, simulation dynamic test is applied in a real case study in a LV grid of a city in the North of Brazil, for performance analyses of the proposed algorithm. Stored data about users consumption and grid feeders were used to its simulation and analyze.

- Application of a new method of choosing single-phase consumer units for the load balancing process based on the unbalance levels and future states of load consumption, resulting in the efficient attenuation of the load average unbalance between LV-feeders, in comparison to the legacy system method and the bibiographic revision, which consider in both cases only the current load consumption.

Reviewer #3: The abstract is not properly representative of the entire paper. Please make sure that your abstract is properly structured. In some places, the text should be revised to be more clear.

Author’s response:  The abstract has been completely rewritten:

In the new paradigm of urban microgrids load balancing control becomes essential to ensure the balance and quality of energy consumption. Thus, phase load balance method becomes an alternative solution in the absence of the distributed generation sources. Development of efficient and robust load-balancing control algorithms becomes interesting in order to guarantee the load balance between phases and consumers, as well as to establish an automatic integration between the secondary grid and the supervisory center. This article presents a new phase-balancing control model based on Hierarchical Petri nets, to encapsulate procedures, subroutines and to verify the properties of a combined algorithm system, in order to identifies the load unbalance in phases and improve the selecting process of single-phase consumer units for switching, which is based on load unbalance level and its future state of load consumption. Through results is obtained a reliable flow of automated procedures, that effectively guarantees the load equalization in the low-voltage grid.

Reviewer #3: In Section 6, the authors are encouraged to provide a greater depth of discussion (more details), and modify the discussion and conclusion as well.

Author’s response:  This is explained with major details in the attached document. 

Reviewer #3: Figs 1, 2, 3, 4, 5, 6, 7, 8, 9, 10, 13 are not clear. They are very difficult to read. Please rewrite all these figures. Unless the authors can clearly conduct, this paper should not be accepted

Author’s response:  All the figures shown have been reprinted. Please see in the attached document. 

Round  2

Reviewer 1 Report

It is acceptable in present form.

Reviewer 2 Report

the revised paper is fine

Reviewer 3 Report

Thank you for putting the effort to revise the paper.